# DiViNeT: 3D Reconstruction from Disparate Views via Neural Template Regularization

**Aditya Vora**[1]     **Akshay Gadi Patil**[1]     **Hao Zhang**[1,2]

[1]Simon Fraser University      [2]Amazon

## Abstract

We present a volume rendering-based neural surface reconstruction method that takes as few as three *disparate* RGB images as input. Our key idea is to regularize the reconstruction, which is severely ill-posed and leaving significant gaps between the sparse views, by learning a set of *neural templates* to act as surface priors. Our method, coined DiViNet, operates in two stages. It first learns the templates, in the form of 3D Gaussian functions, across different scenes, without 3D supervision. In the reconstruction stage, our predicted templates serve as anchors to help "stitch" the surfaces over sparse regions. We demonstrate that our approach is not only able to complete the surface geometry but also reconstructs surface details to a reasonable extent from a few disparate input views. On the DTU and BlendedMVS datasets, our approach achieves the best reconstruction quality among existing methods in the presence of such sparse views and performs on par, if not better, with competing methods when dense views are employed as inputs.

## 1   Introduction

3D reconstruction from multi-view images is a fundamental task in computer vision. Recently, with the rapid advances in neural fields [51, 31] and differentiable rendering [18], many methods have been developed for neural 3D reconstruction [34, 55, 23, 28, 48, 22, 54, 36, 62, 50]. Among them, volume rendering-based methods have achieved impressive results. Exemplified by neural radiance fields (NeRF) [31], these approaches typically employ compact MLPs to encode scene geometry and appearance, with network training subjected to volume rendering losses in RGB space.

The main drawback of most of these methods is the requirement of dense views with considerable image overlap, which may be impractical in real-world settings. When the input views are sparse, these methods often fail to reconstruct accurate geometries due to radiance ambiguity [61, 49]. Despite employing smoothness priors during SDF/occupancy prediction as an inductive bias [34, 55, 48, 54], the limited overlap between regions in the input images causes little-to-no correlation to the actual 3D surface, resulting in holes and distortions, among other artifacts, in the reconstruction.

Several recent attempts have been made on neural reconstruction from sparse views. SparseNeuS [25] learns generalizable priors across scenes by constructing geometry encoding volumes at two resolutions in a coarse-to-fine and cascading manner. While the method can handle as few as 2 or 3 input images, the corresponding views must be sufficiently close to ensure significant image overlap and quality reconstruction. MonoSDF [58] relaxes on the image overlap requirement and relies on geometric monocular priors such as depth and normal cues to boost sparse-view reconstruction. However, such cues are not always easy to obtain, e.g., MonoSDF relied on a pre-trained depth estimation model requiring 3D ground-truth (GT) data. Moreover, depth prediction networks often rely on cost volumes [52, 14] which are memory expensive and thus restricted to predicting low-resolution depth maps. Because of this, the results obtained typically lack fine geometric details [58].

37th Conference on Neural Information Processing Systems (NeurIPS 2023).

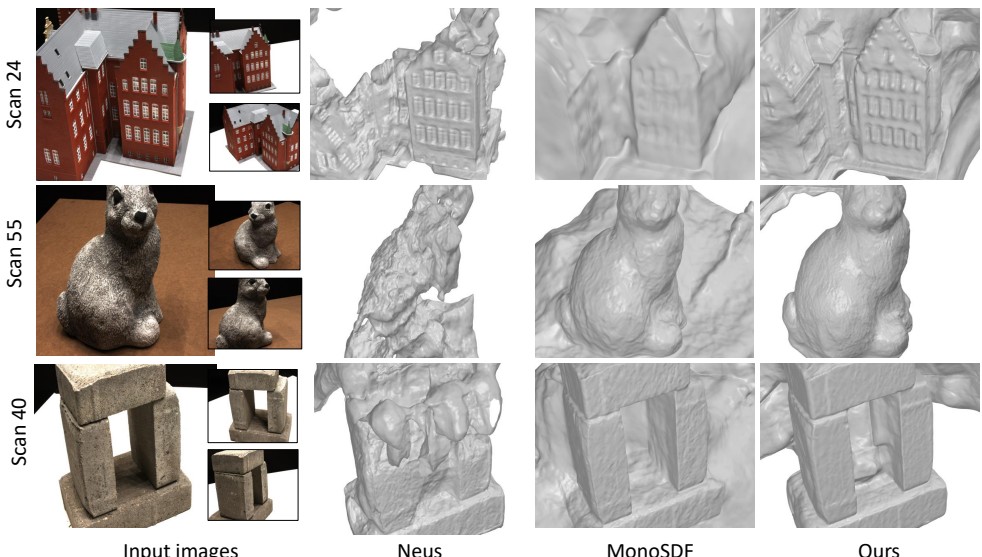

Figure 1: Surface reconstruction results on the DTU dataset [16] for three disparate view inputs (shown on the left). We compare our technique against NeuS[48], which does not use any regularization, and against MonoSDF [58], which uses additional depth and normal supervision to regularize the surface reconstruction process. The results clearly show that our method not only reconstructs the complete surface but also preserves sharp geometric details *without* using any other ground truth information beyond the RGB inputs. See Section 4.1 for a detailed explanation.

In this paper, we target neural surface reconstruction from sparse *and disparate* views. That is, not only are the input RGB images few in number, they also do no share significant overlap; see Figure 1. Our key idea is to regularize the reconstruction, which is severely ill-posed and leaves significant gaps to be filled in between the few disparate views, by learning a set of *neural templates* that act as surface priors. Specifically, in the first stage, we learn the neural templates, in the form of 3D Gaussian functions, across different scenes. Our network is a feedforward CNN encoder-decoder, trained with RGB reconstruction and auxiliary losses against the input image collection. In the second stage, surface reconstruction via volume rendering, our predicted templates serve as anchors to help "stitch" the surfaces over sparse regions without negatively impacting the ability of the signed distance function (SDF) prediction network from recovering accurate geometries.

Our two-stage learning framework (see Figure 2) is coined DiViNet for neural 3D reconstruction from Disparate Views via NEural Templates regularization. We conduct extensive experiments on two real-world object-scenes datasets, viz., DTU [16] and BlendedMVS [53], to show the efficiency of our method over existing approaches on the surface reconstruction task, for both sparse and dense view input settings. Through ablation studies, we validate the design choices of our network in the context of different optimization constraints employed during training.

## 2   Related Work

**View-based 3D Reconstruction.** Reconstructing 3D surfaces from image inputs is a non-trivial task mainly due to the difficulty of establishing reliable correspondences between different regions in the input images and estimating their correlation to the 3D space. This is especially pronounced when the number of views is *limited* and *disparate*. Multi-view stereo (MVS) techniques, both classical [1, 3, 6, 9, 44, 12, 4, 41, 45], and learning-based [27, 47, 59, 39, 15, 52, 57], while using an explicit 3D representation, address the problem of multi-view 3D reconstruction either by estimating depth via feature matching across different input views or by voxelizing 3D space for shape reconstruction.

A recent exploration of neural implicit functions for 3D shape/scene representation [29, 37, 30, 24, 46, 40, 17] has led to an explosion of 3D reconstruction from multi-view images, where two prominent directions exist – one based on surface rendering techniques [34, 55, 23], which require accurate 2D masks at the input, and the other based on volume rendering techniques [28, 48, 22, 54, 36, 62, 50],

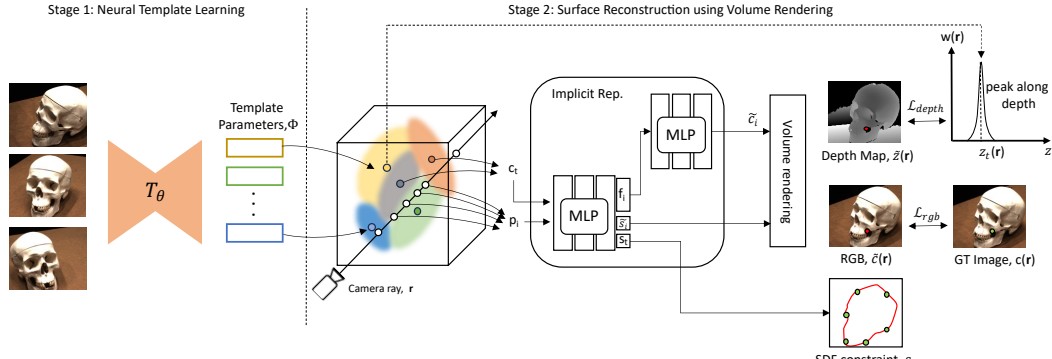

Figure 2: A two-stage learning framework developed to reconstruct 3D surfaces from sparse, disparate view input images. In Stage 1 (see Section 3.1), we train a network across different scenes to predict structured templates (Gaussians, in our case) that encode surface priors. In Stage 2 (see Section 3.2), we train a surface reconstruction model to reconstruct the surface SDF values by leveraging the predicted templates from Stage 1. The purpose served by the predicted templates is in aiding the reconstruction process by acting as regularizers, allowing to obtain complete geometry (see Figure 1 and 7), and even details to a reasonable extent (see Figure 5), from disparate view inputs.

which require no such object masks. The latter techniques combine the representational power of neural implicit with the recently proposed NeRF models [31], achieving better surface reconstructions over the former techniques. A major drawback of these works is that they all require *dense* multi-view images as input. Two recent techniques, Sparse NeuS [25] and Mono-SDF [58] learn 3D reconstruction from just three input images. In the case of [25], the three input images have a significant amount of overlap, and can not be employed when the views are disparate. [58], on the other hand, leverages additional cues such as ground truth depths and normals, allowing reconstruction from disparate views. Our work differs from these in the sense that we do not make use of any explicit cues in the form of ground truth depths and normals, while still being able to faithfully reconstruct 3D surfaces from just three disparate RGB images as input.

**Regularizing Neural Surface Reconstruction.** Our ability to reconstruct 3D surfaces from sparse, disparate view images is made possible through template-based regularization that provides helpful surface priors during the volume rendering stage. Regularization has been explored in the context of novel-view synthesis [33, 7], where patch-based regularization for both the scene geometry and color is performed [33] or additional cues such as depth supervision are incorporated [7] while handling sparse inputs in both cases. In the context of surface reconstruction, MonoSDF [58] uses depth and normal supervision as a part of unsaid regularization to obtain surface geometry from sparse input images. Other works [5, 60, 8] study the effect of regularization on the volumetric rendering-based surface reconstruction framework. Their goal is to improve surface reconstruction by imposing multi-view photometric and COLMAP [43] constraints on the Signed Distance Field (SDF) prediction during training. Such methods show significant improvements over vanilla approaches for surface reconstruction. However, under sparse scenarios, because of significant view differences, photometric and COLMAP constraints are difficult to satisfy, resulting in poor reconstruction quality. In contrast to these methods, inspired by [11, 10, 35], we propose to learn surface priors in the form of templates across a data distribution and use these templates to guide the reconstruction process.

## 3   Method

Our method achieves 3D reconstruction of solid objects from a small set of RGB images $\{I_i\}_{i=0}^{N-1}$ with very less visual overlap, where $N$ can be as less as 3 images and $I_i \in [0,1]^{H \times W \times 3}$. We assume that camera extrinsics and intrinsics are known for all the images. As shown in Figure 2 our approach is realized in two stages: (1) Learning a network for predicting shape templates (see Section 3.1), and (2) Leveraging the predicted templates, learning a volumetric surface reconstruction network with depth and SDF constraints (see Section 3.2).

### 3.1 Stage 1: Learning Surface Priors with Neural Templates

**Template Representation.** We represent the shape with a bunch of $N_t$ local implicit functions called templates, where the influence of $i^{th}$ template is represented as $g_i(p, \varphi_i)$ and is given by,

$$g_i(p, \varphi_i) = s_i \exp \left( \sum_{d \in \{x,y,z\}} \frac{-(c_{i,d} - p_d)^2}{2r_{i,d}^2} \right), \quad (1)$$

where $p \in \mathbb{R}^3$ is the 3D location of a point in the volume and $\varphi_i$ is the template parameter. Specifically, each template $\varphi_i$ is represented as a scaled, anisotropic 3D Gaussian, $\varphi_i \in \mathbb{R}^7$, which consists of a scaling factor $s_i$, a center point $c_i \in \mathbb{R}^3$, and per-axis radii $r_i \in \mathbb{R}^3$. Using this local implicit function, we can find the probability of a point near the surface by summing up the influences of all $N_t$ templates, i.e. $G(p, \Phi) = \sum_{i=1}^{N_t} g_i(p, \varphi_i)$, where $\Phi$ is the set of all the template parameters.

**Network Architecture.** Figure 3 shows the template training stage. Given a fixed set of $N_t$ templates ($\Phi$) and input images set $\{I_i\}$, we aim to optimize a network $\mathcal{T}_\theta$ for the task of template prediction which comprises of an encoder ($\mathcal{E}$) and decoders ($\mathcal{D}_{geo}, \mathcal{D}_{vox}$). The encoder is a feed-forward Convolutional Neural Network (CNN), which we first use to extract 2D image features from individual input views. We then obtain per template latent code $z_{\varphi_i} \in \mathbb{R}^{64}$ using a bi-linear interpolation step as shown in Figure 4. For this step, depending on the number of templates,

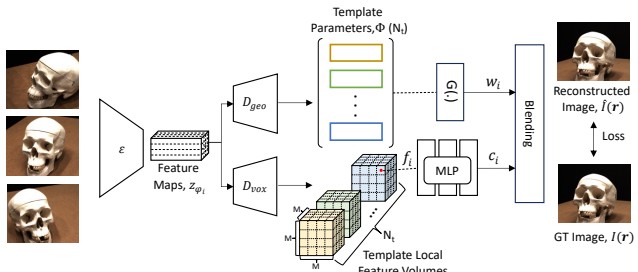

Figure 3: Template Learning stage. We train the template prediction network across scenes to predict $N_t$ structured templates $\Phi$, alongwith local feature volumes per template.

we create a uniform grid of points, which we then use for the interpolation of features obtained from convolutional layers. Each latent code of the template, $z_{\varphi_i}$ is then decoded through two decoders $\mathcal{D}_{geo}$ and $\mathcal{D}_{vox}$. $\mathcal{D}_{geo}$ is the geometry decoder with simple MLP layers predicting the template parameters $\varphi_i$. That is, $\mathcal{D}_{geo} : \mathbb{R}^{64} \to \mathbb{R}^7$. $\mathcal{D}_{vox}$, on the other hand, is a feature decoder comprised of transposed convolutional layers. The objective of $\mathcal{D}_{vox}$ is to map the template feature representation, $z_{\varphi_i}$, to a local volumetric feature grid for each individual template.

Decoding all the template latent codes $z_{\varphi_i}$ at once through $\mathcal{D}_{vox}$ results in a dense voxel grid $V \in \mathbb{R}^{(\mathcal{C}*\mathcal{M}) \times (\mathcal{M}*\sqrt{N_t}) \times (\mathcal{M}*\sqrt{N_t})}$, which we then rearrange to $N_t$ local volumes of size $\mathbb{R}^{\mathcal{C} \times \mathcal{M}^3}$, where $\mathcal{C}$ is the feature dimension of each voxel and $\mathcal{M}$ is the resolution of the feature grid. In our experiments, we set $N_t = 576$, $C = 8$ and $\mathcal{M} = 16$. For more details refer to supplementary.

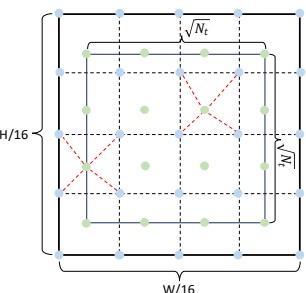

Figure 4: Interpolation step. $\sqrt{N_t} \times \sqrt{N_t}$ codes obtained from $H/16 \times W/16$ features.

**Network Training.** Due to the lack of ground truth 3D information of objects in the training set, we use RGB reconstruction loss along with auxiliary losses to optimize the template parameters. Given the predicted feature volume from the template network, we can query a continuous field at any 3D query point $p_i$ by passing its tri-linearly interpolated feature $f_{vox}(p_i)$, along with the location $p_i$ to a small MLP, consisting of 2 layers and 256 hidden dimensions. The output of the MLP is the color prediction at that particular query point, $p_i$ in the 3D space. In addition to color, we compute each point's blending weights $w_i$ in 3D space using our predicted templates. This blending weight is computed by summing up the local influences of each template on the query point. We then integrate the predicted colors and weights to obtain the predicted image $\hat{I}$. For $K$ query points sampled along a ray, this can be written as:

$$\hat{I}(r) = \sum_{i=1}^{K} w_i c_i, \qquad w_i = \frac{G(p_i, \Phi)}{\sum_i G(p_i, \Phi) + \epsilon} \tag{2}$$

Here, $p_i$ is a query point along a ray, $\epsilon$ is a small perturbation to prevent $w_i$ from diverging to infinity when $G(p_i, \Phi)$ takes small values, whereas $G(p_i, \Phi) = \sum_{i \in N_t} g_i(p_i, \phi_i)$. We then optimize our template prediction network using RGB reconstruction loss. To make sure that the predicted templates are plausible and near the surface, we propose auxiliary loss functions. The total loss by combining all the losses is given as:

$$\mathcal{L}_{total} = \mathcal{L}_{rgb} + \lambda_{cd}\mathcal{L}_{chamfer} + \lambda_c\mathcal{L}_{cov} + \lambda_r\mathcal{L}_{radius} + \lambda_v\mathcal{L}_{var} \tag{3}$$

Here, $\mathcal{L}_{rgb}$ is the mean square error loss between the predicted pixel $\hat{I}(\mathbf{r})$ and ground truth pixels $I(\mathbf{r})$ which is defined as:

$$\mathcal{L}_{rgb} = \sum_{r \in \mathcal{R}} \|\hat{I}(r) - I(r)\|_2^2 \tag{4}$$

$\mathcal{L}_{chamfer}$ is the chamfer distance loss, which minimizes the distance between the template centers $\{C_t : c_1, c_2, \cdots, c_{N_t} \in \mathbb{R}^3\}$, with the corresponding sparse reconstructed point cloud $\{S_c : s_1, s_2, \cdots, s_N \in \mathbb{R}^3\}$ obtained using triangulation, from COLMAP[43]. This loss makes sure that the predicted templates from the network are near the surface. This is given by:

$$\mathcal{L}_{chamfer} = \frac{1}{|C_t|} \sum_{c_i \in C_t} \min_{s_j \in S_c} \left( \|c_i - s_j\|_2^2 \right) + \frac{1}{|S_c|} \sum_{s_j \in S_c} \min_{c_i \in C_t} \left( \|s_j - c_i\|_2^2 \right) \tag{5}$$

$\mathcal{L}_{cov}$ is the coverage loss, which is defined over the entire sparse points, $S_c$. This ensures that the predicted templates cover the entire object. It is defined as,

$$\mathcal{L}_{cov} = \frac{1}{|S_c|} \sum_{s \in S_c} \left(1 - \sigma\left(\sum_{i=1}^{N_t} g_i(s, \varphi_i)\right)\right) \tag{6}$$

where $\sigma(.)$ is the sigmoid function. In addition to this, we also use radius $\mathcal{L}_{radius}$ and variance loss $\mathcal{L}_{var}$, which penalizes large and skewed template radii. This is given by:

$$\mathcal{L}_{radius} = \sum_{r_i \in R_t} \|r_i\|_2^2, \qquad \mathcal{L}_{var} = \sum_{r_i \in R_t} \|r_i - \bar{r}_t\|^2 \tag{7}$$

Here, $R_t$ is the set of all template radii. $r_i$ is the individual template radii, and $\bar{r}_t$ is the mean radius across all the templates, $N_t$.

### 3.2 Stage 2: Surface Reconstruction using Volume Rendering

**Geometry Representation and Volume Rendering.** Following the recent works of surface reconstruction with volume rendering [48, 54, 36], we represent the volume contained by the object to be reconstructed using a signed distance field $\tilde{s}_i$, which is parameterized using an MLP $f_{\theta_s}$, with learnable parameters $\theta_s$ i.e $\tilde{s}_i = f_{\theta_s}(\gamma(x_i))$. Here $\gamma(.)$ is the positional encoding of point in 3D space [31]. Given the signed distance field, the surface $\mathcal{S}$ of the object is represented as its zero-level set i.e.

$$\mathcal{S} = \{x \in \mathbb{R}^3 | f_{\theta_s}(x) = 0\}. \tag{8}$$

To train the SDF network in volume rendering framework, we follow [48], and use the below function, to convert the SDF predictions, $f_{\theta_s}(p_i)$ to $\alpha_i$ at each point, $p_i$ in the 3D space.

$$\alpha_i = \max\left(\frac{\Phi_s(f_{\theta_s}(p_i)) - \Phi_s(f_{\theta_s}(p_{i+1}))}{\Phi_s(f_{\theta_s}(p_i))}, 0\right). \tag{9}$$

Here, $\Phi_s(x) = (1 + e^{-sx})^{-1}$ is the *sigmoid* function and $s$ is learned during training.

Along with the signed distance field, $\tilde{s}_i$, we also predict color, $\tilde{c}_i$ for each sampled point in 3D space using a color MLP, $f_{\theta_c}$. Following [55] the input to the color MLP is a 3D point $p_i$, a viewing

direction $\hat{d}$, analytical gradient $\hat{n}_i$ of our SDF, and a feature vector $z_i$ computed for each point $p_i$ by $f_{\theta_s}$. Hence, the color MLP is the following function, $\tilde{c}_i = f_{\theta_c}(p_i, \hat{d}, \hat{n}_i, z_i)$. We optimize these networks using volume rendering, where the rendered color of each pixel is integrated over $M$ discretely sampled points $\{p_i = o + t_i\hat{d} | i = 1, \cdots, M, t_i < t_{i+1}\}$ along a ray traced from camera center $o$ and in direction $\hat{d}$. Following [31] the accumulated transmittance $T_i$ for a sample point $p_i$ is defined as $T_i = \prod_{j=1}^{i-1}(1 - \alpha_j)$, where $\alpha_j$ is the opacity value defined in Eq. 9. Following this, we can compute the pixel color and depth as follows:

$$\hat{C}(r) = \sum_{i=1}^{M} T_i \alpha_i \tilde{c}_i, \qquad\qquad \hat{z}(r) = \sum_{i=1}^{M} T_i \alpha_i t_i \qquad (10)$$

**Optimization.** The overall loss function we use for optimizing for the surface is:

$$\mathcal{L} = \mathcal{L}_{color} + \lambda_1 \mathcal{L}_{eikonal} + \lambda_2 \mathcal{L}_{depth} + \lambda_3 \mathcal{L}_{sdf} \qquad (11)$$

$\mathcal{L}_{color}$, is the L1 loss between the predicted and ground truth colors which is given as:

$$\mathcal{L}_{color} = \sum_{r \in \mathcal{R}} \|\hat{C}(r) - C(r)\|_1 \qquad (12)$$

Following [13] we use Eikonal loss to regularize the SDF in 3D space, which is given as:

$$\mathcal{L}_{eikonal} = \sum_{x \in \mathcal{X}} (\|\nabla f_{\theta_s}(x)\|_2 - 1)^2 \qquad (13)$$

In addition to these, we propose the following regularization terms in order to aid the surface reconstruction process, especially in sparse reconstruction scenarios.

**Depth Loss**: We sample depth cues $z_t(r)$ along each ray using the templates predicted by our template network. These depth cues are obtained by finding the depth locations where the template weight function $w(x)$ at a point $x$, mentioned in 2 attains a maximum value among all the points sampled along a ray. We then minimize the L1 loss $\mathcal{L}_{depth}$ between the predicted depth, $\hat{z}(r)$ and the computed depth cues $z_t(r)$ as:

$$\mathcal{L}_{depth} = \sum_{r \in \mathcal{R}_{valid}} \|\hat{z}(r) - z_t(r)\|_1 \qquad (14)$$

Here, $\mathcal{R}_{valid}$ denotes the set of rays that intersect the templates in the 3D space. We find these rays using ray sphere intersection.

**SDF Loss**: In addition to depth, we also constrain our SDF to pass through the template centers which we assume are near the surface. This is done by minimizing the L1 loss between the SDF at the template centers and the zero-level set. Here, $N_t$ is the number of template parameters.

$$\mathcal{L}_{sdf} = \frac{1}{N_t} \sum_{x \in C_t} \|f_{\theta_s}(x)\|_1 \qquad (15)$$

**Implementation Details.** Our implementation is based on the PyTorch framework [38]. For both stages, we use the Adam optimizer [20] to train our networks and set a learning rate of $5e^{-4}$. For stage-1, we set $\lambda_{cd}$, $\lambda_c$, $\lambda_r$ and $\lambda_v$ to 1.0, 0.1, 0.1, and 1.0, respectively. And for stage-2, we set $\lambda_1$, $\lambda_2$ and $\lambda_3$ to 1.0, 0.8 and 0.8, respectively. As the templates are not perfect, we follow [58] and use an exponentially decaying loss weight for both SDF and depth regularization for the first $25k$ iterations of optimization. During training in both stages, we sample a batch of 512 rays in each iteration. During both stages, we assume that the object is within the unit sphere. Our SDF network $f_{\theta_s}$, is an 8 layer MLP with 256 hidden units with a skip connection in the middle. The weights of the SDF network are initialized by geometric initialization [2]. The color MLP is a 4 layer MLP with 256 hidden units. The 3D position is encoded with 6 frequencies, whereas the viewing direction is

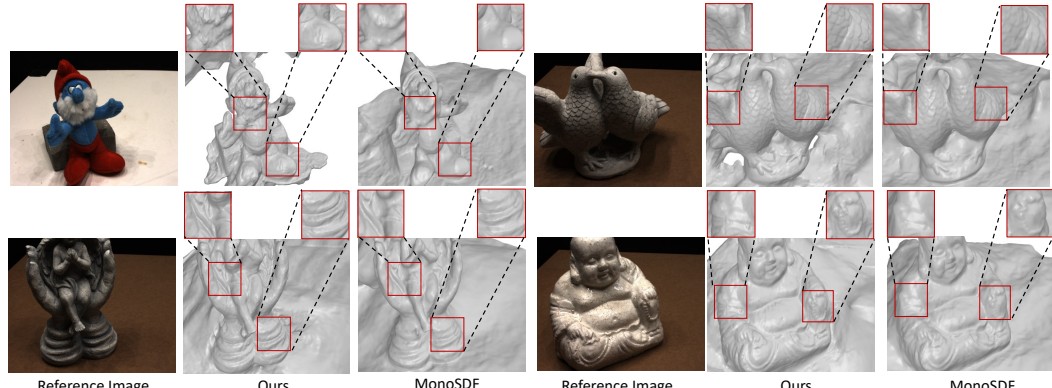

| Reference Image | Ours | MonoSDF | Reference Image | Ours | MonoSDF |

Figure 5: This figure compares the details on the reconstructed geometry, for sparse-view inputs on the DTU dataset, against the second-best competing method, MonoSDF (see Table 1). Our approach produces sharper details, which can be attributed to the surface guidance from the predicted templates.

encoded with 4 frequencies. We train the network for 300k iterations which takes roughly 8 hours on NVIDIA RTX 3090Ti GPU. After training, we extract the mesh using marching cubes [26] at a $512^3$ resolution. See the Supplementary material for more details.

## 4 Results and Evaluation

**Datasets.** We use two commonly employed real-world object-scenes datasets, viz., DTU [16] and BlendedMVS dataset [53]. DTU dataset contains multi-view images (49-64) of different objects captured with a fixed camera and lighting parameters. For training sparse view reconstruction models, we use the same image IDs as in [33, 58, 56]. In addition to this, in order to show the generalization ability of our template prediction network we experiment with our method on the MobileBrick dataset [21], which consists of high-quality 3D models along with precise ground-truth annotations.

For the template learning stage, our training split consists of objects that do not overlap with the 15 test scans of the DTU dataset. We provide the list of scans used for training in the supplementary. We evaluate our method on standard 15 scans used for evaluation by [34, 55, 54, 48, 36]. The resolution of each image in the dataset is $1200 \times 1600$. Unlike MonoSDF [58] which resizes the images to $384 \times 384$ resolution for training, we input the actual image resolution to our pipeline. The other dataset, BlendedMVS, consists of 113 scenes captured from multiple views, where each image is of resolution $576 \times 768$. Unlike DTU, BlendedMVS dataset has not been employed before in the sparse view setting, and as such, specific view IDs to train and test are not available. We therefore manually select 3 views that have little overlap.

**Baselines.** For sparse views, we compare against the classical MVS method, COLMAP [42], and recent volume rendering-based methods such as NeuS [48], VolSDF [54] and MonoSDF [58]. We exclude SparseNeuS [25] evaluation in disparate settings because of its inability to reconstruct 3D geometry when the input images have significantly less overlap. For dense views, we additionally compare against the recent grid-based method, Voxurf [50]. A mesh from COLMAP point cloud output is obtained using Screened Poisson Surface Reconstruction [19].

**Evaluation Metrics.** Following standard evaluation protocol, we use Chamfer Distance (CD) between the ground truth point cloud and predicted meshes to quantify the reconstruction quality. On the DTU dataset, following prior works [54, 48, 58], we report CD scores. On the BlendedMVS dataset, we simply show qualitative results as is commonly done in the literature. In addition to this, for evaluation on the MobileBrick dataset, we use F1 score computed using the ground-truth mesh.

### 4.1 Results on DTU

Figure 1 and Table 1 show qualitative and quantitative results, respectively, in the presence of sparse view inputs (three, to be specific) on the DTU dataset. We observe from Table 1 that our method outperforms all other competing methods, including MonoSDF [58], on nine of the fifteen scenes in the test set, and achieves the lowest mean CD score (the next best adds an additional 0.09 CD score).

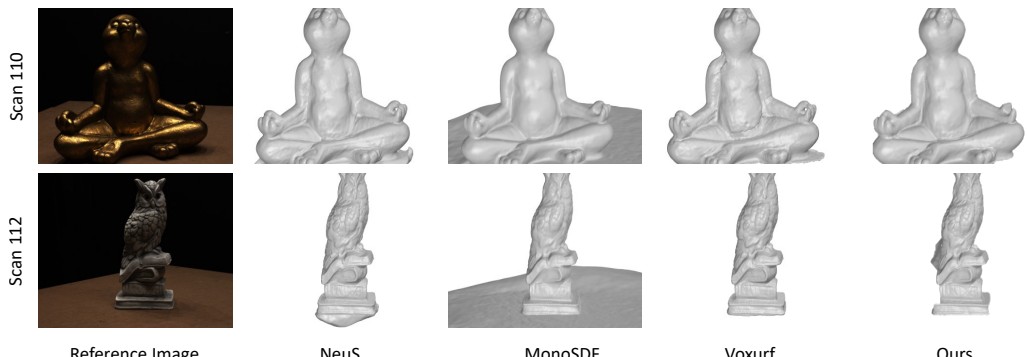

Figure 6: Dense view reconstruction results on the DTU dataset.

Table 1: Quantitative comparisons for *sparse and disparate* view 3D reconstruction on DTU dataset (Section 4.1). For Scan ID 110, × indicates that COLMAP could not reconstruct any points.

| | Chamfer Distance (CD) (↓) | | | | | | | | | | | | | | | |
|---|---|---|---|---|---|---|---|---|---|---|---|---|---|---|---|---|
| Scan ID → | 24 | 37 | 40 | 55 | 63 | 65 | 69 | 83 | 97 | 105 | 106 | 110 | 114 | 118 | 122 | Mean |
| COLMAP[41] | 4.45 | 4.67 | 2.51 | 1.9 | 2.81 | 2.92 | 2.12 | 2.05 | 2.93 | 2.05 | 2.01 | × | 1.1 | 2.72 | 1.64 | 2.56 |
| VolSDF [54] | 5.24 | 5.09 | 3.99 | 1.42 | 5.1 | 4.33 | 5.36 | 3.15 | 5.78 | 2.07 | 2.79 | 5.73 | 1.2 | 5.64 | 6.2 | 4.2 |
| NeuS [48] | 4.39 | 4.59 | 5.11 | 5.37 | 5.21 | 5.60 | 5.26 | 4.72 | 5.91 | 3.97 | 4.02 | 3.1 | 4.84 | 3.91 | 5.71 | 4.78 |
| MonoSDF [58] | 3.47 | **3.61** | 2.1 | 1.05 | **2.37** | **1.38** | 1.41 | 1.85 | **1.74** | 1.1 | 1.46 | **2.28** | 1.25 | 1.44 | **1.45** | 1.86 |
| Ours | **3.37** | 4.11 | **1.46** | **0.75** | 2.74 | 1.52 | **1.13** | **1.63** | 2.08 | **0.98** | **0.87** | 2.7 | **0.47** | **1.24** | 1.57 | **1.77** |

On the other six scans, our method is the second-best performing method behind MonoSDF. This is because our predicted templates are not always perfect for all objects, meaning, regions on the surface with no predicted templates potentially suffer from incomplete reconstructions. Other methods such as VolSDF [54] and NeuS [48] fail to even beat the classical MVS method, COLMAP [43], indicating that these methods either require dense view inputs[54] or require a significant amount of overlap on the input views [25] to faithfully reconstruct the surface. This emphasizes the effectiveness of our method on the disparate views case. Further, Figure 5 compares surface details on the reconstructed geometry against, MonoSDF [58], where it can be observed that our method produces sharper details, attributed mainly to the geometry guidance from learned template priors.

On dense view inputs (49-64), Figure 6 and Table 3 present qualitative and quantitative results, respectively, over the aforementioned techniques. We additionally compare against a recent method, Voxurf [50]. In this setting, all neural surface reconstruction

Table 2: Generalization of TPN on MobileBrick dataset.

| | F1 Score (↑) | | | |
|---|---|---|---|---|
| Object → | Bridge | Camera | Colosseum | Castle |
| Ours (pre-trained TPN) | 0.565 | 0.61 | 0.219 | 0.175 |
| Ours (TPN re-training) | **0.658** | **0.67** | **0.22** | **0.187** |

methods outperform COLMAP, indicating that the presence of dense view inputs is necessary to tap into the learning ability of neural rendering techniques for improved geometry reconstruction. We observe that our method outperforms the rest on five scans, which is higher than any other method (Voxurf-three scans, NeuS-four, MonoSDF-two, and VolSDF-one). Overall, our method achieves the second-best mean CD score, closely trailing behind Voxurf (a difference of 0.03 CD score). This is expected as feature-grid-based approaches tend to reconstruct sharper details compared to MLP [58]. But otherwise, we beat all other existing works owing to our neural template regularization scheme. More qualitative results on dense view inputs are provided in the Supplementary material.

### 4.2 Results on BlendedMVS

We also run our pipeline on a more challenging, also real-world, BlendedMVS dataset [53]. We use the sparse input setting and show comparisons against the same set of prior works, i.e., NeuS[48] and MonoSDF [58]. Figure 7 shows qualitative results on 4 object-scenes with complex geometries. Clearly, our method obtains the best surface reconstruction. NeuS is not able to reconstruct the

Table 3: Quantitative comparisons for *dense-view* 3D reconstruction on DTU dataset (Section 4.1).

| | Chamfer Distance (CD) (↓) | | | | | | | | | | | | | | | |
|---|---|---|---|---|---|---|---|---|---|---|---|---|---|---|---|---|
| Scan ID → | 24 | 37 | 40 | 55 | 63 | 65 | 69 | 83 | 97 | 105 | 106 | 110 | 114 | 118 | 122 | Mean |
| COLMAP[41] | 0.81 | 2.05 | 0.73 | 1.22 | 1.79 | 1.58 | 1.02 | 3.05 | 1.4 | 2.05 | 1.0 | 1.32 | 0.49 | 0.78 | 1.17 | 1.36 |
| NeuS[48] | 1.0 | 1.37 | 0.93 | 0.43 | 1.1 | **0.65** | **0.57** | 1.48 | **1.09** | 0.83 | **0.52** | 1.2 | 0.35 | 0.49 | 0.54 | 0.84 |
| VolSDF [54] | 1.14 | 1.26 | 0.81 | 0.49 | 1.25 | 0.7 | 0.72 | **1.29** | 1.18 | 0.7 | 0.66 | 1.08 | 0.42 | 0.61 | 0.55 | 0.86 |
| MonoSDF [58] | 0.83 | 1.61 | 0.65 | 0.47 | **0.92** | 0.87 | 0.87 | 1.3 | 1.25 | **0.68** | 0.65 | 0.96 | 0.41 | 0.62 | 0.58 | 0.84 |
| Voxurf [50] | 0.91 | **0.73** | **0.45** | **0.34** | 0.99 | 0.66 | 0.83 | 1.36 | 1.31 | 0.78 | 0.53 | 1.12 | 0.37 | 0.53 | 0.51 | **0.76** |
| Ours | **0.78** | 1.29 | 0.68 | 0.42 | 0.96 | 0.70 | 0.68 | 1.43 | 1.30 | 0.827 | 0.62 | **0.94** | **0.34** | **0.49** | **0.49** | 0.79 |

Figure 7: Qualitative reconstruction results for three disparate view inputs on the challenging BlendedMVS dataset [53]. See Section 4.2 and the Supplementary for a more details explanation.

geometry faithfully for any of the objects due to disparate view inputs. MonoSDF, on the other hand, despite using ground truth depth and normal cues, fails to reconstruct accurate geometry, especially for Clock and Sculpture objects. In contrast to this, our method can accurately reconstruct the geometric details owing to template guidance coming from Stage 1.

## 4.3 Generalization on MobileBrick Dataset

We test the generalization ability of our template prediction network (TPN) on a new dataset, MobileBrick [21], by comparing the reconstruction results when the neural templates were learned by a pre-trained TPN (on DTU) vs. when they were trained on MobileBrick. Note that overall, the models from these two datasets are quite different in terms of geometry and structure. The results in Table 2 and qualitative results in Figure 8a (a) show that the reconstruction qualities under the two scenarios are comparable, attesting to the generalizability of our TPN. As mentioned before, the metric used in this evaluation is F1 score as reported by MobileBrick dataset.

## 4.4 Ablation Studies

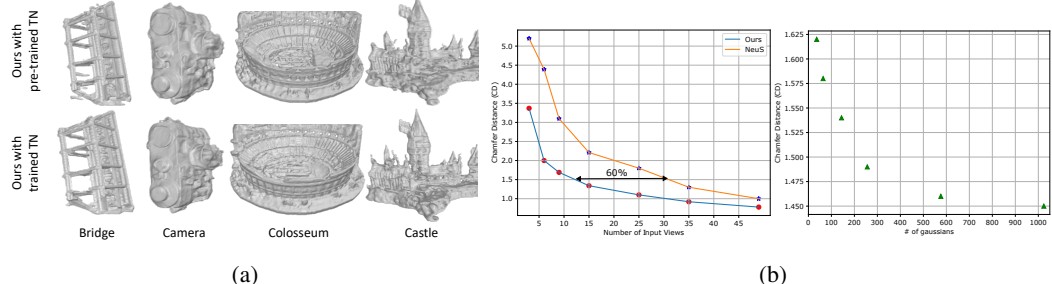

(a)                                           (b)

Figure 8: (a) Generalization of TPN on MobileBrick Dataset. (b) Effect on chamfer distance with # of Views (left) and Gaussians (right).

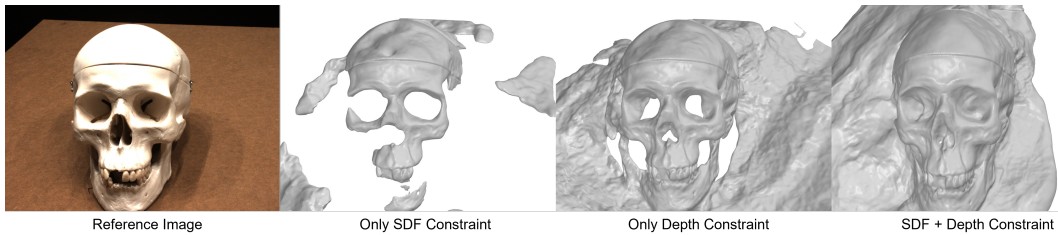

Reference Image          Only SDF Constraint          Only Depth Constraint          SDF + Depth Constraint

Figure 9: Effect of different optimization constraints (Figure 2, Section 3.2) employed in our pipeline.

In Figure 9, we demonstrate the need for employing different loss functions as described in Section 3.2 and depicted on the right side of Figure 2 and Table 4. We observe how the reconstructed geometry changes when $\mathcal{L}_{sdf}$ and $\mathcal{L}_{depth}$ are separately employed. Evidently, only SDF constraints cannot reconstruct surfaces due to the coarse supervision from template centers. On the other hand, depth regularization significantly improves the reconstruction but holes

Table 4: Impact of each constraint.

|  | CD ($\downarrow$) |
|---|---|
| Only SDF Constraint | 2.70 |
| Only Depth Constraint | 1.96 |
| **SDF + Depth Constraint** | **1.52** |

still exist. Our full pipeline includes both the above constraints, resulting in improved reconstruction, as can be seen in the last column of Figure 9 and Table 4. We also study reconstruction quality as a function of the number of input views. Figure 8b depicts the behavior of two such methods i.e. NeuS and Ours, up to fifty views. Our method achieves the same Chamfer Distance (CD) score for twelve views as NeuS achieves for around thirty. This demonstrates the efficacy of our method to "catch" surface geometry even when sufficient view inputs are not present. In addition to this, we measure the effect on reconstruction quality by varying the number of Gaussians. As it can be seen in Figure 8b the surface reconstruction quality significantly improves as we increase the number of gaussians. The reconstruction quality saturates after 576 gaussians and hence we use 576 gaussians throughout our experiments.

## 5    Discussions, Limitations and Future Work

We present DiViNet for sparse multi-view 3D reconstruction. Our core idea is a simple two-stage framework with neural Gaussian templates learned across different scenes serving as surface priors to regularize the volume-rendering-based reconstruction. Experiments demonstrate quality reconstructions by DiViNet in both dense and sparse+disparate view settings, with clearly superior performance over state-of-the-art in the latter.

One current limitation of our method is its reconstruction speed. While comparable to NeuS [48], VolSDF [54], and MonoSDF [58], it is still slow due to the two-stage approach, especially when generalizing to a large number of real-world scenes. In addition to addressing these issues, e.g., dynamic templates, we would also like to replace the MLPs in DiViNet with more efficient grid-based techniques, e.g., instant NGP [32], and explore more general primitives such as convexes and super-quadrics in place of Gaussians.

# 6 Acknowledgements

We thank the anonymous reviewers for their valuable comments and Sherwin Bahmani and Yizhi Wang for discussions during rebuttal and helping with the renderings in the paper. This research is supported in part by a Discovery Grant from NSERC (No. 611370).

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
