# DiViNeT: 3D Reconstruction from Disparate Views via Neural Template Regularization -Supplementary Material-

**Aditya Vora**[1]    **Akshay Gadi Patil**[1]    **Hao Zhang**[1,2]

[1]Simon Fraser University    [2]Amazon

## A   Architecture & Implementation Details

We present a novel approach for a volume rendering-based surface reconstruction, which takes as few as 3 images as input. Below are the details of each step.

### A.1   Template Prediction network architecture.

Inspired by the generalization approaches of neural radiance field [1, 19, 14], we take a similar approach for our template learning step. Our network $\mathcal{T}_{\theta_0}$ is composed of an encoder ($\mathcal{E}$) and 2 decoders i.e. geometry decoder ($\mathcal{D}_{geo}$), and voxel feature-grid decoder ($\mathcal{D}_{vox}$).

**Encoder** ($\mathcal{E}$)**.**  The encoder ($\mathcal{E}$) is adapted from [15].  There are a total of 5 convolutional layers in the encoder, which downsamples the image by a factor of 32. Hence, with $N$ images as input at a resolution of $512 \times 512 \times 3$, we obtain a feature of size $N \times C \times 16 \times 16$, post the forward pass. Here, $C$ is the number of feature channels obtained at the output ($C = 64$). Using these features obtained from the encoder, we sample per template feature using bilinear interpolation. We concatenate all the interpolated template features $\{f_i\}_{i=1}^{\mathcal{N}_t}$ into a tensor of size $\mathcal{N}_t \times N \times C$.

We aggregate the multi-view features of each template, obtained from images using the feature-pooling strategy proposed in [12]. This allows us to input any number of images as input. Specifically, we first compute the element-wise mean, $\mu$, and variance, $v$ for each feature obtained from the encoder $\mathcal{E}$. We then concatenate each feature with the computed, $\mu$ and $v$ and feed it through the shared MLP followed by a softmax to predict the weight $\{w_i\}_{i=1}^{N}$ for each multi-view feature. We then blend all the features from all views using the predicted blend weights $\{w_i\}_{i=1}^{N}$. Post this step, the template features $\{f_i\}_{i=1}^{\mathcal{N}_t}$ gets transformed into a tensor of size $\mathcal{N}_t \times C$.

**Decoders** ($\mathcal{D}_{geo}$ **and** $\mathcal{D}_{vox}$)**.** $\mathcal{D}_{geo}$ is used to predict the template parameters, whereas $\mathcal{D}_{vox}$ predicts voxel-grid features for each template.  We decode these features channel-wise into a feature of dimension $C \cdot \mathcal{M} \times \sqrt{\mathcal{N}_t} \cdot \mathcal{M} \times \sqrt{\mathcal{N}_t} \cdot \mathcal{M}$. We then reshape these features to a volumetric feature grid of dimension $C \times \mathcal{M} \times \sqrt{\mathcal{N}_t} \cdot \mathcal{M} \times \sqrt{\mathcal{N}_t} \cdot \mathcal{M}$. Here, $C = 8$, $\mathcal{M} = 16$. The architectures are illustrated in Table 1 2.

### A.2   Training Details.

To learn generalizable templates, we train our template prediction network end-to-end across different objects from DTU [5] and BlendedMVS datasets [16]. For the DTU dataset, we use 15 scenes for testing, which are the same scenes used by [18]. The remaining non-overlapping 75 scenes are used for training our template prediction network. This split is the same split that is used by [8].

The BlendedMVS [16] dataset, contains $> 100$ scenes, captured in both indoor and outdoor settings. This includes many scenes with complex architecture and backgrounds. We exclude these complex

37th Conference on Neural Information Processing Systems (NeurIPS 2023).

Table 1: The architecture of the two decoders, $\mathcal{D}_{geo}$ and $\mathcal{D}_{vox}$. Each template feature is, $f_i$. A hidden feature of each template is, $h_i$. $[C_t, R_t, S_t]$ is the set of centers, radii, and scales of all the templates. L=Linear, R=ReLU, TCN=Transposed Convolution. The volumetric feature grid of each template is of size $16 \times 16 \times 16$. We use a total of $576(24^2)$ templates.

| Decoder | Layer | Channels | Input | Output |
|---|---|---|---|---|
| $\mathcal{D}_{geo}$ | $LR_{[0-3]}$ | 64 / 64 | $f_i$ | $h_i$ |
| | $L_{ctr}$ | 64 / 3 | $h_i$ | $C_t$ |
| | $L_{rad}$ | 64 / 3 | $h_i$ | $R_t$ |
| | $L_{scale}$ | 64 / 1 | $h_i$ | $S_t$ |
| $\mathcal{D}_{vox}$ | $TCN_0$ | 64 | $24 \cdot 1 \times 24 \cdot 1 \times 64$ | $24 \cdot 2 \times 24 \cdot 2 \times 64$ |
| | $TCN_1$ | 32 | $24 \cdot 2 \times 24 \cdot 2 \times 64$ | $24 \cdot 4 \times 24 \cdot 4 \times 32$ |
| | $TCN_2$ | 64 | $24 \cdot 4 \times 24 \cdot 4 \times 32$ | $24 \cdot 8 \times 24 \cdot 8 \times 64$ |
| | $TCN_3$ | $16 \times 8$ | $24 \cdot 8 \times 24 \cdot 8 \times 64$ | $24 \cdot 16 \times 24 \cdot 16 \times 16 \cdot 8$ |

Table 2: Architecture of the MLP used to regress colors, $\mathbf{c}$ at point locations $\mathbf{p}$ in 3D space. $f_{vox}(\mathbf{p})$ is the per-point feature obtained from tri-linear interpolation.

| Layer | Channels | Input | Output |
|---|---|---|---|
| Reshape | - | $24 \cdot 16 \times 24 \cdot 16 \times 16 \cdot 8$ | $24 \cdot 16 \times 24 \cdot 16 \times 16 \times 8$ |
| $LR_0$ | 3 + 8 / 256 | $\mathbf{p}$, $f_{vox}(\mathbf{p})$ | hidden feature |
| $LR_1$ | 256 / 3 | hidden feature | $\mathbf{c}$ |

large-scale scenes in our template training. Out of all the objects, we select 35 objects for training the template prediction network and, following [17, 11, 8] we select 7 other non-overlapping objects for the test split which we use to train our surface reconstruction model.

### A.3 Sparse Reconstruction using COLMAP.

Before the template training stage, we reconstruct a sparse point cloud for all the objects in our train split using COLMAP [10]. This point cloud acts as a "free" supervision [2, 14] for the optimization of our template prediction network. To reconstruct the point cloud, we run COLMAP in triangulation mode, with all the provided images and ground truth camera poses of objects in the train split. We use these sparse reconstructed point clouds to optimize our template prediction network.

## B  Additional Results

### B.1  BlendedMVS Dataset

We show additional results on the BlendedMVS dataset for 3 new objects. As can be in Figure 2 our reconstruction quality significantly outperforms NeuS [11] and MonoSDF [20]. This shows that our method is capable to do surface reconstruction from disparate views even in complex settings. In contrast to this, MonoSDF with MLP fails to reconstruct the geometry faithfully. Whereas, NeuS [11] without any regularization has holes in the reconstruction. We show the images used to train the surface reconstruction model alongside the reconstruction results.

### B.2  DTU Sparse Views

Figure 1 shows additional qualitative comparison results on DTU scans with 3 views as input. We compare our method with NeuS [11] and MonoSDF [20]. As can be seen, our method achieves superior reconstruction quality compared to both NeuS and MonoSDF. We are able to achieve this, without using any explicit ground truth information. In contrast to this, MonoSDF uses pixel-accurate depth and normal map predictions in order to regularize the surface reconstruction process in sparse view settings.

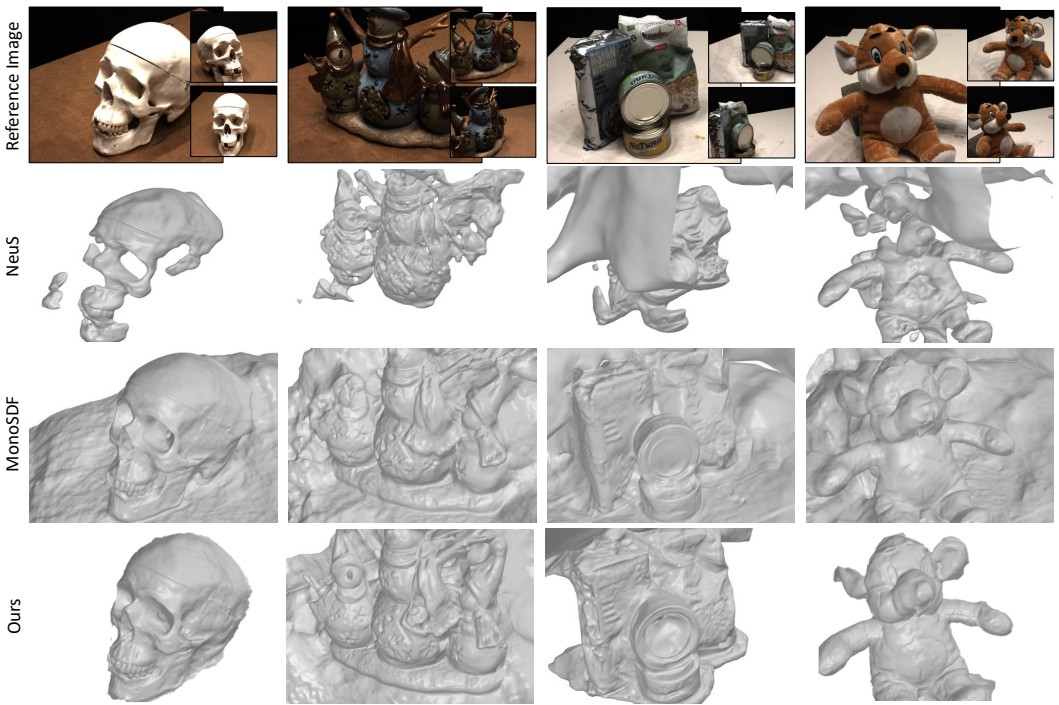

Figure 1: Results on DTU dataset with 3 views.

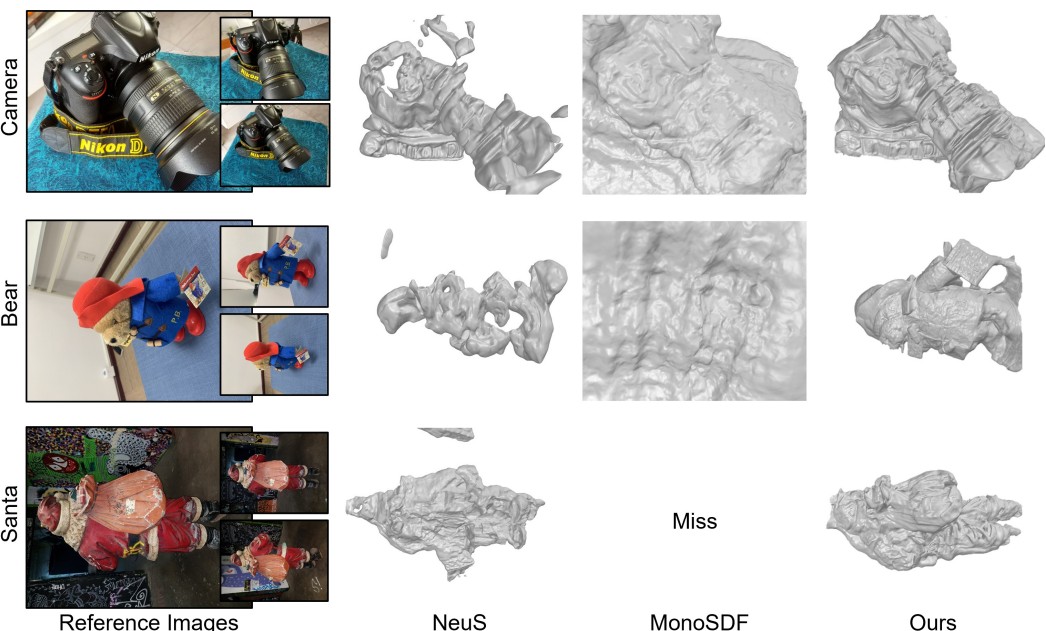

Figure 2: Results on BMVS dataset with 3 views. "Miss" represents the absence of reconstruction because of the "Out of Memory" issue.

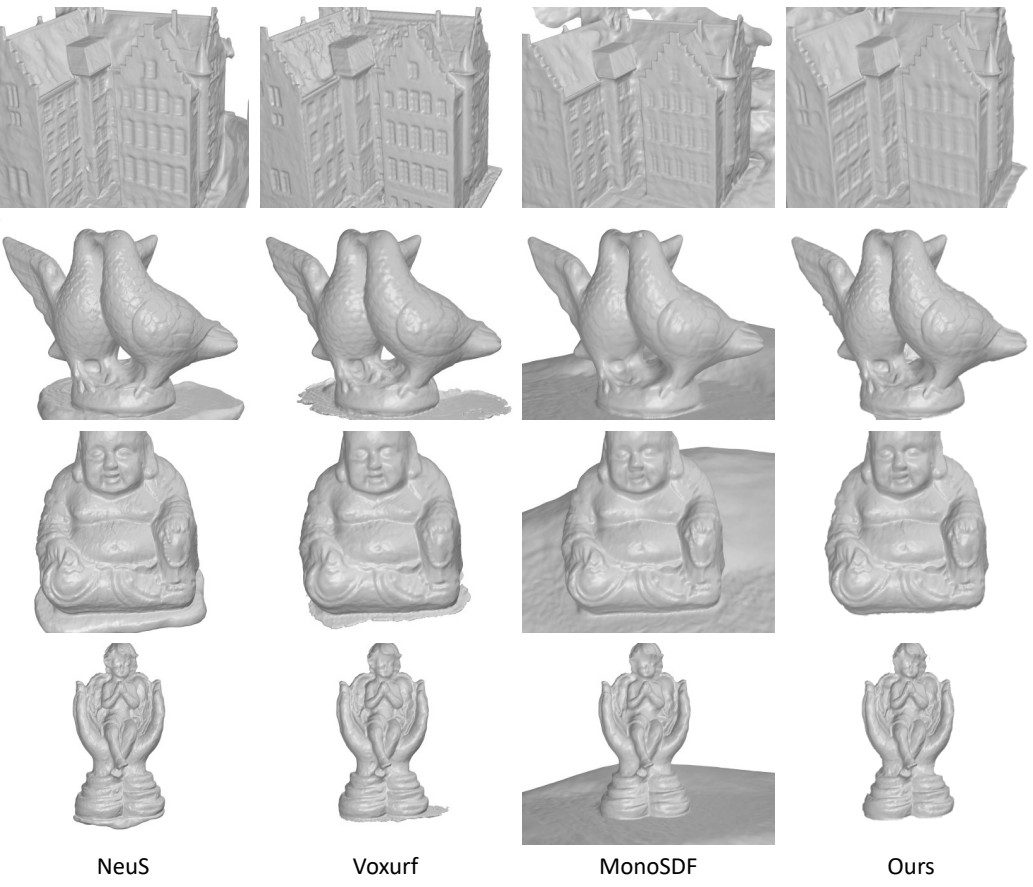

|  NeuS | Voxurf | MonoSDF | Ours |

Figure 3: Results on DTU dataset with dense views.

Table 3: Quantitative Comparison on Mobilebrick dataset with GeoNeuS.

| | F1 Score (↑) | | | |
|---|---|---|---|---|
| Object → | Bridge | Camera | Colosseum | Castle |
| GeoNeuS [3] | 0.74 | 0.73 | 0.36 | 0.457 |
| Ours | **0.915** | **0.846** | **0.415** | **0.572** |

## B.3 DTU Dense Views

Figure 3 shows additional qualitative comparison results on DTU scans with dense views as input. We compare our method against [11, 20, 13] for dense views. As it can be seen our method can achieve equally good reconstruction quality when dense views are provided as input.

## B.4 Additional Comparisons on MobileBrick Dataset

In addition, we also compare our method against GeoNeus [3] which uses a sparse reconstructed point cloud from COLMAP [9] and photometric consistency to regularize the surface reconstruction from dense views. In the case of scenes that are captured in the wild, COLMAP reconstruction is often noisy. Using this point cloud for regularization can negatively affect the reconstruction as observed in [7]. In contrast to this, DiViNet uses template priors which are trained across data. Such data-driven priors have been shown to be immune to outliers and noise [4]. We validate this by comparing our results with GeoNeus on dense views on the MobileBrick dataset [6]. The qualitative results are shown in Figure 4 and quantitative results are shown in Table 3.

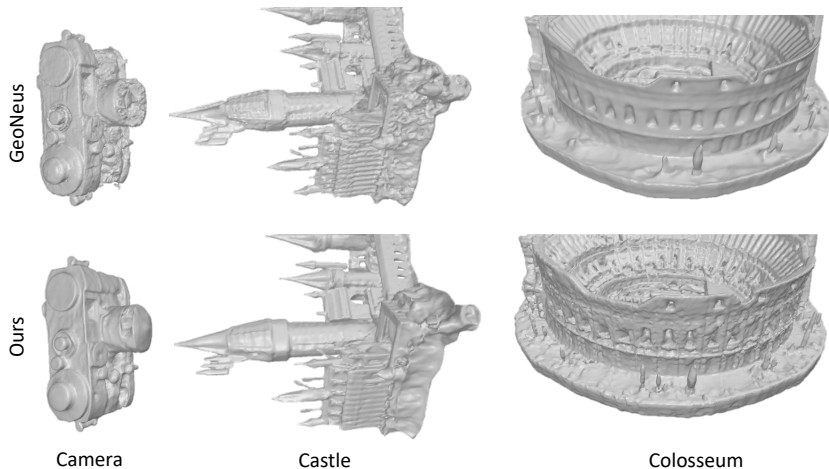

Figure 4: Qualitative Comparison with GeoNeus.

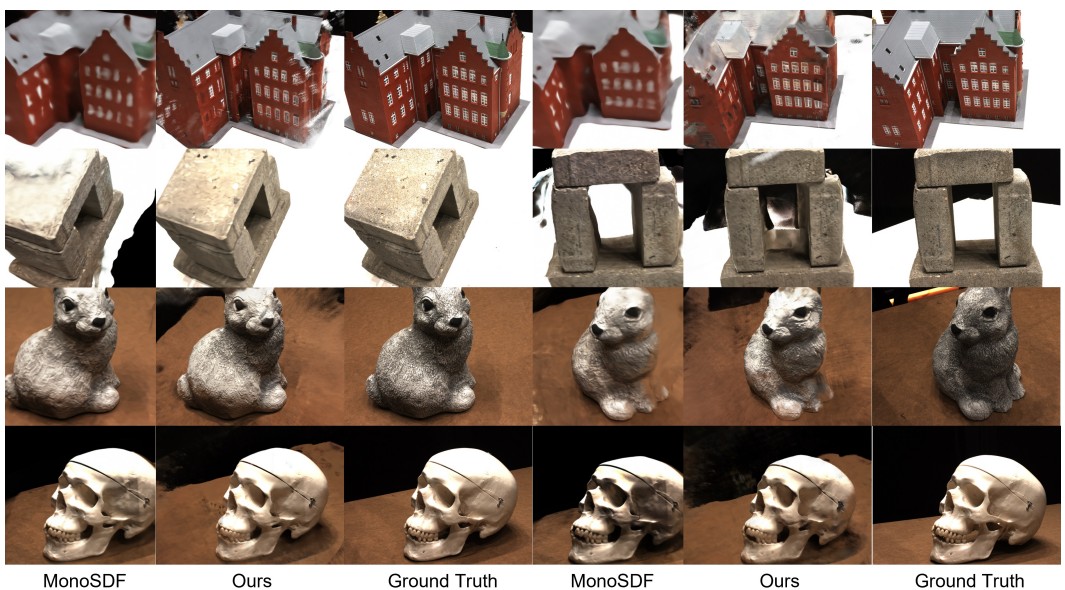

Figure 5: Novel View Synthesis Results on DTU dataset trained using 3 views.

## B.5 Novel View Synthesis

Figure 5, 6, 7 shows the novel view synthesis results of different objects from the DTU dataset. As can be seen from the figures, our method achieves better view synthesis quality compared to MonoSDF on unseen views. Table 4 shows the quantitative results.

Table 4: Quantitative results of Novel View Synthesis.

|  | PSNR ($\uparrow$) | SSIM ($\uparrow$) | LPIPS ($\downarrow$) |
| --- | --- | --- | --- |
| MonoSDF [20] | 23.64 | — | — |
| Ours | **24.34** | **0.7208** | **0.264** |

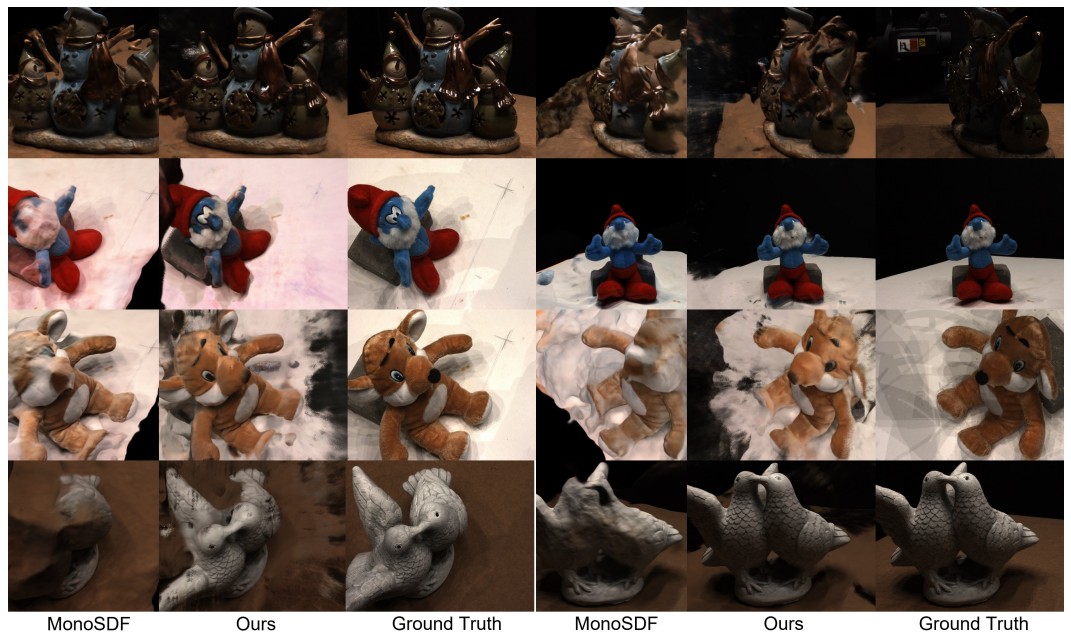

| MonoSDF | Ours | Ground Truth | MonoSDF | Ours | Ground Truth |

Figure 6: Novel View Synthesis Results on DTU dataset trained using 3 views.

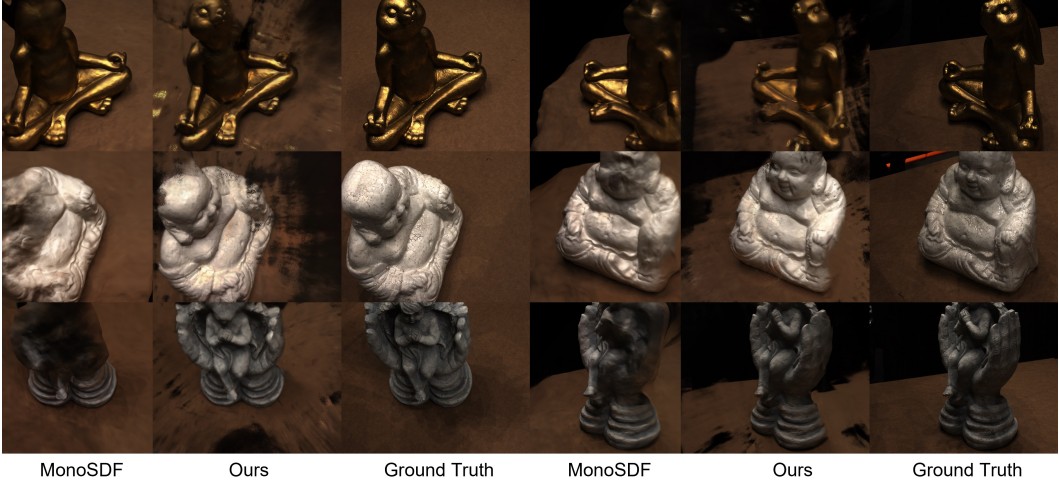

| MonoSDF | Ours | Ground Truth | MonoSDF | Ours | Ground Truth |

Figure 7: Novel View Synthesis Results on DTU dataset trained using 3 views.

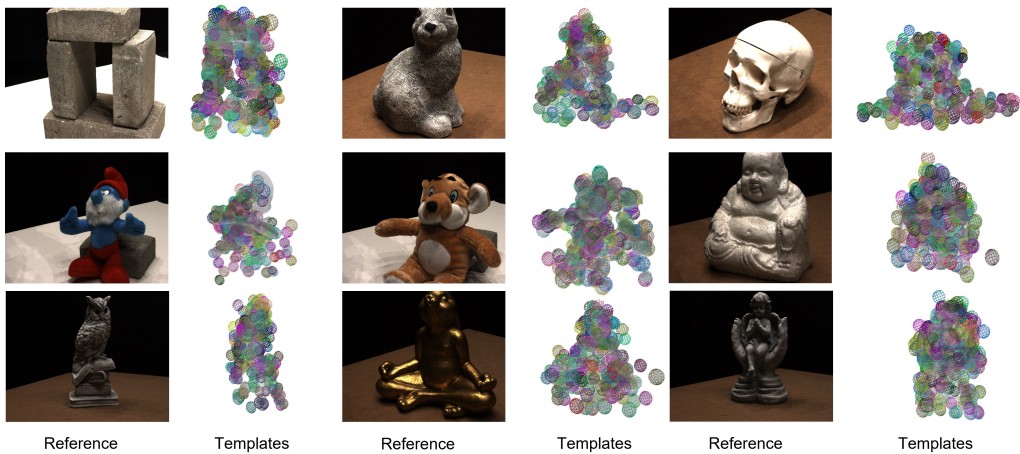

| Reference | Templates | Reference | Templates | Reference | Templates |

Figure 8: Template predictions on DTU dataset.

### B.6 DTU Template Predictions

Figure 8 shows our template predictions on the DTU objects on the test split. During the inference of our templates on the test split, it is possible that the templates contain outliers i.e. those whose centers are not near the surface and have skewed radii. Hence, we do an outlier removal step, where we remove the templates whose centers and radii are 2 standard deviation away from the mean center and radii of templates respectively. In Figure 8 we show results after the outlier removal step.