# OpenReview forum: "DiViNeT: 3D Reconstruction from Disparate Views using Neural Template Regularization"
_NeurIPS.cc/2023/Conference — NeurIPS 2023 poster_

### Official Review · Reviewer_bjQu · 2023-06-30

**Soundness:** 3 good
**Presentation:** 2 fair
**Contribution:** 2 fair
**Rating:** 4
**Confidence:** 4

**Summary:**

This paper proposes a two-stage framework for neural 3D reconstruction from disparate views via neural templates regularization. In the first stage, a network is trained for predicting shape templates. After that the volumetric surface reconstruction network with depth and SDF constraints is trained with the templates prior. Compared to other 3D reconstruction methods, DiViNet specifically targets disparate input images and achieves SOTA in sparse image views.

**Strengths:**

1.The proposed DiViNet gets rid of the require of explicit cues, unlike former methods. Moreover, only a small set of images with small overlaps are used.

2.This proposed designs are technically sound.

**Weaknesses:**

1. Although DiViNet performs best with sparse view inputs, with dense view inputs, it's worse than the latest methods. According to Table 2, DiViNet is the second-best method.
The DiViNet achieves SOTA in too many constraints to other methods. MonoSDF[1] using multi-resolution feature grids gets 0.73 on DTU dataset.

2. As mentioned in the paper, the neural templates lacks generalization in different data distribution. It matters how big the influence of the data distribution is. The authors claim the drawback of other methods is the requirement of dense views with overlap. The generalization to new data distribution is also important in consideration of real-world setting.

[1]MonoSDF: Exploring Monocular Geometric Cues for Neural Implicit Surface Reconstruction


**Questions:**

Please refer to Weaknesses.

**Limitations:**

Please refer to Weaknesses.

---

> ### Author Rebuttal · Authors · 2023-08-08
>
> Thank you for the valuable comments.
>
> > **Q1 - Performance on dense-view scenarios**
>
> As addressed in Q2 in the global response and in the paper, our current implementation uses MLPs for the reconstruction task, and as such, for a fair comparison, we compare against the MLP representation of MonoSDF which achieves a CD score of 0.84. However, there was an apparent boost in CD score in MonoSDF by simply replacing the MLPs with a multi-resolution hash grid, as in [1]. We expect a similar performance boost in our method with this replacement. We are happy to provide such results in the revision. Furthermore, we include a dense view comparison to GeoNeuS in Q2 in the global response. We show a significant margin to GeoNeuS.
>
> > **Q2 - Generalization of template learning network**
>
> This concern has been addressed in Q1 of the global response.
>
> [1] Müller, Thomas, et al. "Instant Neural Graphics Primitives with a Multiresolution Hash Encoding.", ToG 2022.

---

> ### Comment · Reviewer_bjQu · 2023-08-21
>
> Thank the authors for the response.
>
> After reading the rebuttal and other reviews, I will keep my original rating.

---

### Official Review · Reviewer_4rRP · 2023-07-06

**Soundness:** 3 good
**Presentation:** 3 good
**Contribution:** 4 excellent
**Rating:** 7
**Confidence:** 4

**Summary:**

This paper proposes DiViNet for sparse multi-view reconstruction, which specifically targets on sparse input as few as three disparate RGB images. The key is to regularize the reconstruction process by learning a set of neural templates as surface priors, which is basically a set of 3D gaussian functions with optimizable features.  Extensive experiments are conducted to demonstrate the quality of DiViNet in sparse and disparate view settings.

**Strengths:**

•	This paper focuses on an important and practice problem. Priors works fail to reconstruct accurate geometries when input views are sparse, while this paper proposes novel neural temporal regularization method to achieve good quality even with only three disparate images as input.
•	The idea of learning surface priors with neural templates is novel, and it’s effectiveness is also validated through the experiment section.
•	Some interesting observations are draw from the experiment part, for example, under sparse view input cases, VolSDF and NeuS even fail to bear COLMAP, while under dense input cases, all the neural methods outperform COLMAP.


**Weaknesses:**

•	The ability of learned neural template to generalize to new scenes is not very clear. On one hand they are learned across different scenes, on the other hand the author claimed the need to be learned again when deployed to datasets from a different data distribution. A clarification and explanation on under what scenario can the learned template functions be reused will be very helpful.
•	Since each learned template is represented as a scaled, anistropic 3D gaussian, a visualization showing the learned gaussian from a scene and the corresponding sparse reconstructed pointcloud from Colmap will be helpful. I am especially curious about the positional distribution and the scale of Nt gaussian, whether the query points are affected by only a small number of 3D gaussians or not.


**Questions:**

•	There are some point-based neural representation works which also consider 3D gaussian as their main geometry primitive, I feel the neural template function in this paper share a lots of insight with those papers. Including those paper in related works and providing some discussion will make the paper more stronger.
o	3D Gaussian Splatting for Real-Time Radiance Field Rendering. Siggraph 2023
o	Neural Point Catacaustics for Novel-View Synthesis of Reflections. ToG 2022


**Limitations:**

The authors have discussed potential negative social impacts of this paper.

---

> ### Author Rebuttal · Authors · 2023-08-08
>
> Thank you for the valuable comments.
>
> > **Q1 - Generalization ability of the Template Prediction Network**
>
> This has been addressed in Q1 in the global response.
>
> > **Q2 - Visualization of learned templates and COLMAP reconstruction**
>
> Please see Fig 4 in the rebuttal pdf for the visualization of the learned templates and COLMAP reconstructed point cloud. Also, we have included many more renderings of templates with the meshes in the supplementary (Fig. 7).
>
> > **Q3 - Related Works**
>
> Thanks for the pointers. We will discuss relevant works in the revised version.

---

### Official Review · Reviewer_Ahim · 2023-07-06

**Soundness:** 2 fair
**Presentation:** 2 fair
**Contribution:** 2 fair
**Rating:** 3
**Confidence:** 4

**Summary:**

This paper addresses the problem of surface reconstruction from spase input views. The authors adopt a SDF representation parameterized by an MLP. They propose learning a set of neural templates (in the form of 3D Gaussian functions) to serve as anchors in the reconstruction process to help stitch the surfaces over sparse regions. Reconstruction is carried out by optimizing the SDF (MLP) through minimizing the rendering loss and the Eikonal loss. The authors introduce a depth loss and SDF loss, both computed based on the estimated neural templates, to regularize the optimzation. Experimental results on the DTU and BlendMVS datasets show the proposed approach can reconstruct surface details to a reasonable extent from few disparate input views.

**Strengths:**

+ The template prediction network can be trained end-to-end using RGB reconstruction loss without any 3D supervision.
+ The predicted templates, which approxmiate points on object surface, help to regularize the SDF optimization, allowing the proposed approach to produce more complete reconstructions with reasonable surface details.
+ The depth loss, which computes the difference between the rendered depth and depth cues obtained from the predicted templates, and the SDF loss, which evaluates the signed distance at template centers, both sound logical and correct in regularizing the SDF optimization. Their effectiveness has been validated through ablation study.

**Weaknesses:**

- It is not clear about the generalization ability of the template prediction network. No training details have been provided in the paper. In the evaluations, are the same datasets being used in both training and testing the template prediction network? What would be the performance if the template prediction network trained on one dataset is applied to a completely different dataset? How will this (negatively) affect the reconstruction?
- Quite often the reconstructions also include incorrect background surface. Is that caused by incorrect template predictions? No discussions or analysis have been included in the paper.
- The description of the template prediction network architecture is very confusing (both in the main paper and the supplementary material). For instance, it is not clear how per-template feature are being sampled using bilinear interpolation from the extracted feature maps. There is also no explanations for the design of the dimension (C x M x sqrt(N_t) . M x sqrt(N_t) . M) for the volumetric feature grid.
- The formulation in (2) may be problematic. Note a ray will in general intersect an object surface in at least 2 points. This implies more than one template will produce a large value for w_k and therefore colors at multiple surface points will be mixed (as depth is not being considered).

**Questions:**

Please refer to the points in weaknesses.
- L_{rgb} in (3) should be L_{color} instead.
- \epsilon in (2) is undefined.
- \sigma in (9) is undefined. How is \sigma and the signed distance being related?


**Limitations:**

Yes

---

> ### Author Rebuttal · Authors · 2023-08-08
>
> Thank you for the valuable feedback.
>
> > **Q1 - Generalization ability of the template prediction network (TPN)**
>
> Please refer to Q1 in the global response for the generalization of the template prediction network.
>
> > **Q2 - Training details**
>
> In addition to the main paper, all the training details have been provided in the supplementary material. We would be happy to add missing training details if specified by the reviewer.
>
> > **Q3 - Evaluation of the TPN**
>
> For evaluating the TPN, we have a separate test set. Every time the TPN is trained, the training and test set, which are non-overlapping, are part of one dataset (e.g., either DTU or Blended MVS, accordingly).
>
> > **Q4 - Background surface in the reconstructed results**
>
> Note that this is not specific to, nor a limitation of our work since all the previous works such as NeuS, VolSDF, MonoSDF, etc. output such backgrounds. This is a common occurrence in this line of work. That said, we agree that since the TPN is not trained with pixel-accurate ground truth cues such as depth and normal maps (the priors in MonoSDF), there will always be imperfections in the predictions of templates (as can be seen in Fig. 4 of the rebuttal pdf). This results in some additional reconstructions in the background. But this can be easily avoided by using masks from pre-trained segmentation models like Segment-Anything [1] and incorporating it in the loss function using a binary cross entropy loss as shown in [2]. We will add this discussion to the revision of the paper.
>
> > **Q5 - Description of the TPN**
>
> We realize some confusion exists in the description of the TPN, which shall be written better in the revised version.
>
> Regarding the per-template feature: Given a fixed number of templates that we would like to predict, we create a uniform grid of pixel locations and use that grid to interpolate features using bilinear interpolation. Hence, all the template parameters are regressed from fixed anchors in the feature grid. With this process, we get the latent codes for each template which are then decoded using $D_{geo}$ and $D_{vox}$ decoders.
>
> > **Q6 - The formulation in (2) is problematic**
>
> A careful examination reveals that there is no problem with Equation (2). The weight function computed from Gaussians is a weighted sum of the local influences of all predicted Gaussians. And since we use a max operator, we get a single peak along a ray whose depth we use to regularize the surface.
>
> [1] Kirillov, Alexander, et al. "Segment Anything.", ICCV 2023 \
> [2] Siddiqui, Yawar, et al. "Panoptic Lifting for 3D Scene Understanding with Neural Fields.", CVPR 2023

---

> > ### Comment · Reviewer_Ahim · 2023-08-17
> >
> > Thank the authors for providing their responses. My concerns on the generalization and evaluation of the TPN are mostly resolved. I still have a few questions regarding TPN:
> > 1. In l120, why is the voxel grid feature V_k indiced by k?
> > 2. In l124, what is the dimension of tri-linearly interpolated feature f_{vox}(p)? Is it C x N_t? Is the tri-linear interpolation performed on the MxMXM grid?
> > 3. In l140, is the sparse point cloud reconstructed from 3 views? How many points are reconstructed and used on average? In that case, the point cloud only covers the visible surface, right? This is in fact related to my question on the formulation of (2). If templates are only predicted for the one side of the object, then there should be only a single peak along a ray. On the other hand, if templates are predicted for a closed surafce, then there should be in general at least two peaks along a ray.
> > 4. In L145, L_{var} tends to make the template "spherical". This seems only suitable for very pointed features but may not approximate general / planar surface well.
> > 5. Regarding the per-template features, do you mean that each feature is sampled at a pre-defined grid location on the image feature maps? For the case of 576 temples, the grid used would be 24x24? Since the input images are captured at different viewpoints, aggregating features at the same spatial location would not be very meaningful. Please kindly further clarify this step.
> >
> > Thank you.

---

> > > ### Author Response · Authors · 2023-08-18
> > >
> > > Happy to hear that our rebuttal helped. In regards to your detailed questions please see our point by point responses below.
> > >
> > > > **Q1 - In l120, why is the voxel grid feature V_k indiced by k?**
> > >
> > > We apologise for the confusion and thanks for pointing this out. We realize that, $V_{k}$ should actually be $V$, which we will revise in the next version.
> > >
> > > The encoder encodes the images into a per-template latent code (as described in the response to your Q5), and once these latent codes are obtained, we decode these codes into voxel grid features $V$ through transposed convolutions in $D_{vox}$. You can think of $V$ as $N_{t}$ local volumes each of size $M\times M\times M\times C$ arranged in a *grid* of size $sqrt(N_t) \times sqrt(N_{t})$.
> > >
> > > > **Q2 - In l124, what is the dimension of tri-linearly interpolated feature f_{vox}(p)? Is it C x N_t? Is the tri-linear interpolation performed on the MxMXM grid?**
> > >
> > > If $X = sqrt(N_t) \times M$ and $Y = sqrt(N_{t}) \times M$ and $Z = M$, then tri-linear interpolation takes place on a $X \times Y \times Z$ grid. Hence the dimension of the tri-linearly interpolated feature is just $C$.
> > >
> > > > **Q3 - In l140, is the sparse point cloud reconstructed from 3 views? How many points are reconstructed and used on average? In that case, the point cloud only covers the visible surface, right? This is in fact related to my question on the formulation of (2). If templates are only predicted for the one side of the object, then there should be only a single peak along a ray. On the other hand, if templates are predicted for a closed surafce, then there should be in general at least two peaks along a ray.**
> > >
> > > Following the previous works [1,2,3], during training we use the COLMAP point cloud reconstructed from *dense* views of the objects used for training the TPN. With this approach, the TPN network will learn how to *interpolate* the sparse regions through templates which can then be used for regularization.
> > >
> > > The scenario about closed surface occurs when we have $360&deg;$ object centric scenes, in which we might get 2 peaks if the templates are very sharp around the surface (which is not the case as per the training strategy of TPN). However, this particular case can be easily handled by sorting the peak depths and choosing the peak which is closer to the camera and using the same regularization term proposed in Eq. $14$ of the main paper.
> > >
> > > In addition to this, we would also like to emphasize that these peaks cannot be considered as an *exact* location of a surface but rather the purpose of these peaks is to just serve as anchors to stitch the surface over sparse regions. The rest of the reconstruction is taken care by the volume rendering step. Hence, volume rendering loss along with the regularization tries to find out the most optimum surface under sparse scenarios in our method.
> > >
> > > Moreover, our empirical analysis shows that there is always a single peak along a ray for the objects taken into consideration in the datasets we experimented with (DTU, BMVS) as mentioned in the response above to Q6.
> > >
> > > > **Q4 - In L145, L_{var} tends to make the template "spherical". This seems only suitable for very pointed features but may not approximate general / planar surface well.**
> > >
> > > $L_{var}$ encourages all the patches to be of similar sizes. This along with $L_{rad}$ will prevent the surface to be approximated only using very few *large* templates. Along with this, $L_{var}$ is also required for a more *stable* training of the TPN, because a skewed radius (in one dimension) will lead to NaN during training which the $L_{var}$ prevents. We have a trade-off here between more stable training and the variety of surfaces these templates can model and hence we balance the two. In addition to this, we do agree that this representation is not a *universal* representation for all type of surfaces, however, 3D gaussians are widely adopted representation in computer graphics for shape abstraction [4,5,6] from which we draw inspiration.
> > >
> > >
> > > [1] Roessle, Barbara, et al. "Dense depth priors for neural radiance fields from sparse input views.", CVPR 2022\
> > > [2] Ren, Yufan, et al. "Volrecon: Volume rendering of signed ray distance functions for generalizable multi-view reconstruction.", CVPR 2023\
> > > [3] Long, Xiaoxiao, et al. "Sparseneus: Fast generalizable neural surface reconstruction from sparse views.", ECCV 2022\
> > > [4] Genova, Kyle, et al. "Learning shape templates with structured implicit functions.", CVPR 2019\
> > > [5] Muraki, Shigeru. "Volumetric shape description of range data using “blobby model”." CGI 1991\
> > > [6] Tretschk, Edgar, et al. "Patchnets: Patch-based generalizable deep implicit 3d shape representations." ECCV 2020

---

> > > > ### Author Response · Authors · 2023-08-18
> > > >
> > > > > **Q5 - Regarding the per-template features, do you mean that each feature is sampled at a pre-defined grid location on the image feature maps? For the case of 576 temples, the grid used would be 24x24? Since the input images are captured at different viewpoints, aggregating features at the same spatial location would not be very meaningful. Please kindly further clarify this step.**
> > > >
> > > > Yes, we do use pre-defined grid locations to sample the features to a fixed size based on the number of templates. However, the reason why it works is because of the *view-dependent* feature aggregator [7] which basically does a *weighted pooling* based on the features from different views which contribute towards the template prediction. So if a feature in one image is a good representation of the geometry comprised by a template, that particular feature will be given more weight by the "view-dependent" feature aggregator and the remaining two image features will have less weight. In this way, the network is able to learn to extract the right information for template prediction from images.
> > > >
> > > >
> > > > [7] Wang, Qianqian, et al. "Ibrnet: Learning multi-view image-based rendering." CVPR 2021

---

### Official Review · Reviewer_hUMT · 2023-07-07

**Soundness:** 2 fair
**Presentation:** 3 good
**Contribution:** 3 good
**Rating:** 4
**Confidence:** 4

**Summary:**

This work presents a volume rendering-based sparse view neural surface reconstruction method. For the hard sparse view reconstruction ,the authors propose to learn neural templates as surface priors to guide the learning of neural fields. The results on DTU and Blended MVS are better than NeuS and MonoSDF .

**Strengths:**

The idea of learning templates for enhancing spase view reconstrucion is novel and make sense.


**Weaknesses:**

This paper did not compare with the the most related works on sparse view reconstruction like SparseNeuS and VolRecon. Comparing only with NeuS / MonoSDF are not convincing, since this methods do not have special designes for sparse view reconstruction.

The results on dense view reconstruction are quite poor, shown in Fig. 4. My concern lies in the poor results in dense view reconstruction, i.e. the artifacts in Fig.4, where all the baselines perform better. The quantitative comparisons shown in Tab.2 further deepen my concerns, where the SOTAs this paper not compared already reduce CD to less than 0.55 (e.g. Geo-NeuS), where this paper can only achieves comparable results with NeuS (0.84 vs. 0.79), which was published two years ago.

I do not think that choosing MonoSDF as the main baseline is convincing, since MonoSDF is mainly designed for scene-level reconstruction, where the monolar priors of MonoSDF are not suitable for object level reconstruction (e.g. DTU).


**Questions:**

Why not follow the experimental settings of SparseNeuS(ECCV22)?

Is the proposed method also suitable for scene-level multi-view reconstrucion (e.g. Replica/ScanNet)?

**Limitations:**

See the weaknesses above.

---

> ### Author Rebuttal · Authors · 2023-08-08
>
> Thank you for the valuable comments.
>
> >**Q1 - No comparisons to SparseNeuS**
>
> We reiterate that our framework is designed to excel at reconstruction from sparse (i.e., few in number) *and* wide-baseline/disparate (i.e., little overlap) view images. Due to the latter criterion, we did not show comparisons to SparseNeuS, whose performance hinges on having sufficient overlap between the input views. We contacted the authors of SparseNeuS who confirmed that disparate view scenarios require significant architectural modifications. We did conduct experiments that showed that if the input views were disparate, then SparseNeuS did not converge.
>
> >**Q2 - No comparisons to VolRecon**
>
> In the case of VolRecon, surface reconstruction occurs through point cloud fusion from different views, rather than a single holistic reconstruction, thereby making the reconstruction vulnerable to outliers under disparate scenarios. This can be observed in our quantitative results in the below table and qualitative results in the rebuttal PDF (Fig. 1). Note that the metric used is Chamfer Distance (CD ($\downarrow$)).
>
> | Scan ID $\rightarrow$ |  24  |  37  |  40  |  55  |  63  |  65  |  69  |  83  |  97  |  105 |  106 |  110 |  114 |  118 |  122 | Mean |
> |:---------------------:|:----:|:----:|:----:|:----:|:----:|:----:|:----:|:----:|:----:|:----:|:----:|:----:|:----:|:----:|:----:|:----:|
> |        VolRecon       | 3.59 | 4.16 | 4.12 |  3.2 | 3.56 | 3.76 | 2.46 | 2.44 | 2.57 | 2.66 | 2.75 | 3.75 |  1.6 |  3.0 | 2.16 | 3.05 |
> |          Ours         | **3.37** | **4.11** | **1.46** | **0.75** | **2.74** | **1.52** | **1.13** | **1.63** | **2.08** | **0.98** | **0.87** | **0.87** | **0.47** | **1.24** | **1.57** | **1.77** |
>
>
> > **Q3 - The artifacts in Fig.4, where all the baselines perform better**
>
> It is true that there are some artifacts in the *background* region in the dense view reconstruction. However, the surface reconstruction accuracy of the object is comparable, if not more, to the baselines shown in the figure. This can be verified from the quantitative results of Table 2 in the main paper.
>
>
> > **Q4 - Dense-view results and GeoNeuS**
>
> Please see Q2 in the global response for explanations about dense-view reconstruction results.
>
> GeoNeuS uses photometric consistency and a sparse reconstructed point cloud from COLMAP to regularize the reconstruction. In the case of scenes that are captured in the wild, COLMAP reconstruction is often noisy. Using this point cloud for regularization can negatively affect the reconstruction as observed in [1,2]. In contrast to this, DiViNet uses template priors which are trained across data. Such data-driven priors have been shown to be immune to outliers and noise [3]. We validate this by comparing our results with GeoNeus on dense views on the MobileBrick dataset [4]. The qualitative results are in the rebuttal PDF (Fig 3) and the quantitative results are in the table in the global response Q2.
>
> > **Q5 - Scene-level reconstruction**
>
> Our templates are only trained for object-level reconstruction. We do not currently try to reconstruct indoor scenes. One challenge, as mentioned in response to R#jSFe, is that the template prediction network relies on the COLMAP reconstructed point cloud to effectively learn surface priors. However, indoor scenes comprise large textureless regions because of which COLMAP reconstruction fails, as observed in [4]. Hence, training the template network becomes a challenge. We agree that this is an interesting direction for future work to investigate how Gaussian templates can be used for reconstructing indoor scenes.
>
>
> [1] Zhang, Jingyang, et al. "Critical Regularizations for Neural Surface Reconstruction in the Wild", CVPR 2022 \
> [2] Li, Zhaoshuo, et al. "Neuralangelo: High-Fidelity Neural Surface Reconstruction", CVPR 2023 \
> [3] Huang, Jiahui, et al. "Neural Kernel Surface Reconstruction", CVPR 2023 \
> [4] Wang, Yusen, et al. "Neuralroom: Geometry-constrained Neural Implicit Surfaces for Indoor Scene Reconstruction", arXiv 2022

---

> > ### Comment · Reviewer_hUMT · 2023-08-19
> > **Thanks for the rebuttal.**
> >
> > Thanks for the rebuttal. I am still not convinced by the response of authors to exclude SparseNeuS for comparison. Where are the results that the authors claim to "did conduct experiments that showed that if the input views were disparate, then SparseNeuS did not converge."? In my experiments, the sparseNeuS may produce not good results under little overlap of 3 views but can still converge and will not crash. I think the authors should share the results that "not convege" and anylyse them.
> >
> > Also, it is much more convincing to conduct experiments on both large overlap and little overlap settings, where you can provide a fair comparison with previous works in "large overlap" and also show your special ability in dealing with the new "little overlap" It is not suitable to just creat a new setting with "little overlap" and show no comparisons to the previous method of sparse view reconstruction due to different experiment settings.
> >
> > Still, the authors do not respond to my concern in "I do not think that choosing MonoSDF as the main baseline is convincing, since MonoSDF is mainly designed for scene-level reconstruction, where the monolar priors of MonoSDF are not suitable for object level reconstruction (e.g. DTU)."

---

> > > ### Author Response · Authors · 2023-08-20
> > >
> > > Thank you for the additional questions.
> > >
> > > > **Q1 - I am still not convinced by the response of authors to exclude SparseNeuS for comparison. Where are the results that the authors claim to "did conduct experiments that showed that if the input views were disparate, then SparseNeuS did not converge."? In my experiments, the sparseNeuS may produce not good results under little overlap of 3 views but can still converge and will not crash. I think the authors should share the results that "not convege" and anylyse them.**
> > >
> > > We appreciate your attention to detail. We did conduct experiments with SparseNeuS in the disparate view setting. In the first step, we trained the entire model from scratch in this new setting, using the default hyperparameters. However, during the fine-tuning step what we experienced was that the one-shot output from the trained model outputs a very noisy mesh and then once the iterations of the fine-tuning progress, the mesh vanishes. Since it did not generate any mesh upon finetuning, we decided to not show the results. In retrospect, we should have provided such details instead of simply calling it a non-convergence.
> > >
> > > Since the result was essentially a “failure”, we contacted the first author of the SparseNeuS paper just to be sure. He did kindly reply and remarked, “If your training images are not overlapped, sparseneus won't work, since sparseneus heavily relies on the matching information of overlapping images.”
> > >
> > > At last, please note that our rebuttal did provide a comparison to VolRecon, a follow-up of SparseNeuS which performs better. We believe this comparison is likely more meaningful. Still, if requested, we are happy to show any qualitative results we could obtain from SparseNeuS in the revision.
> > >
> > > > **Q2: Also, it is much more convincing to conduct experiments on both large overlap and little overlap settings, where you can provide a fair comparison with previous works in "large overlap" and also show your special ability in dealing with the new "little overlap" It is not suitable to just create a new setting with "little overlap" and show no comparisons to the previous method of sparse view reconstruction due to different experiment settings.**
> > >
> > > Our work focuses on the reconstruction problem with sparse and disparate input views. Hence, foremost, our experiments were conducted to convince the readers of this, as shown in Table 1 and Figs. 1, 3, and 5.
> > >
> > > For *completeness*, we also provided results and comparisons under the dense view setting, where our method was not the best, but close to being so. Note that we did not make any claim that our method ought to be the best. The main point we wanted to convey is that our method can achieve the best disparate view reconstruction without significantly sacrificing quality when the views are dense (and with large overlaps).
> > >
> > > The reviewer is correct in that the only input setting that we missed is a few views with large overlaps, as in SparseNeuS. But we made no claim over the superiority of our method in this setting either. If there was no intent to convince the reader of this on our part, we do not think there is any unfairness in not providing comparisons to SparseNeuS or other methods in this particular setting. Again, for *completeness*, we can surely provide such comparisons in the supplementary material. Regardless of where our method places, missing such an experiment is inessential and inconsequential to the main selling point of our work.

---

> > > > ### Author Response · Authors · 2023-08-20
> > > >
> > > > > **Q3: Still, the authors do not respond to my concern in "I do not think that choosing MonoSDF as the main baseline is convincing, since MonoSDF is mainly designed for scene-level reconstruction, where the monolar priors of MonoSDF are not suitable for object level reconstruction (e.g. DTU)."**
> > > >
> > > > A direct response to this question was not provided due to its incorrect premise.
> > > >
> > > > Let us direct all reviewers to Section 4.3 and Table 4 of the MonoSDF paper. MonoSDF did exactly an experiment on DTU for object-level reconstruction from sparse (3) views. The authors made an explicit claim on that experiment, which we quote, “When incorporating the (monocular geometric) cues, the results for both (MLP and multires grids) representations are significantly improved.”
> > > >
> > > > Indeed, while these cues/priors in MonoSDF do have certain advantages for the scene-level reconstruction task, especially over textureless regions where depth and surface normal priors prove to be useful [4,5,6] because of their smoothness, at the same time, they also prove to be useful for object-level reconstruction as shown in experiments on DTU for both dense and sparse scenarios (see Sections 4.3 and 4.4 of the MonoSDF paper). In addition to this, depth has been a popular choice of prior under sparse scenarios for both scene and object-level radiance field reconstruction, see [1,2,3] just as a small sampler.
> > > >
> > > > At last, we would like to reiterate from the paper that in order to have a fair evaluation setting we use the *same* setting as used by the previous methods which do sparse view radiance field reconstruction [1,2], i.e. 3 view reconstruction on image IDs [22,25,28] from DTU dataset. We feel that this setting is the *closest common* setting we can have on which we can benchmark our methods along with being the *most standard one* as it is a widely adopted setting for evaluating sparse view *radiance field* reconstruction [1,2,3]. To our knowledge, MonoSDF is the only method, which shows a sparse view *surface reconstruction* of objects in this setting.
> > > >
> > > > Hence, for more reasons than one, we chose MonoSDF in our paper for comparison, and it came out as the second-best performer in Table 1 in our paper.
> > > >
> > > > [1] Niemeyer, Michael, et al. "Regnerf: Regularizing neural radiance fields for view synthesis from sparse inputs.", CVPR 2022 \
> > > > [2] Yang, Jiawei, Marco Pavone, and Yue Wang. "FreeNeRF: Improving Few-shot Neural Rendering with Free Frequency Regularization.", CVPR 2023 \
> > > > [3] Deng, Kangle, et al. "Depth-supervised nerf: Fewer views and faster training for free.", CVPR 2022 \
> > > > [4] Yu, Zehao, et al. "Monosdf: Exploring monocular geometric cues for neural implicit surface reconstruction.", NeurIPS 2022 \
> > > > [5] Wang, Jiepeng, et al. "Neuris: Neural reconstruction of indoor scenes using normal priors.", ECCV 2022 \
> > > > [6] Uy, Mikaela Angelina, et al. "SCADE: NeRFs from Space Carving with Ambiguity-Aware Depth Estimates.", CVPR 2023

---

### Official Review · Reviewer_jSFe · 2023-07-07

**Soundness:** 4 excellent
**Presentation:** 4 excellent
**Contribution:** 4 excellent
**Rating:** 7
**Confidence:** 5

**Summary:**

The authors propose a framework for sparse view 3D reconstruction from disparate views. A two stage approach is presented for reconstruction of a scene from posed sparse images. In the first stage a template is predicted from the sparse images, represented by a number of parametric 3D gaussians. The second stage uses the predicted template to reconstruct the scene from sparse views. An SDF representation is used to represent the geometry. State of the art performance is shown on shape reconstruction and novel view synthesis.

**Strengths:**

1. **Clarity** : The paper is very well written with great attention to detail. Each component is adequately motivated. The approach section is built up in a very methodical and thorough manner.
2. **Reproducibility**: The exact implementation details and training specifics are made clear, aiding in the reproducibility of the proposed approach.
3. **Results**: The qualitative results especially for shape recovery is very compelling, particularly for scenes with wide baselines.
5. **Novelty**: The use of gaussian templates to guide the reconstruction of the scene under sparse setting is a simple and elegant idea that is also easy to incorporate into existing frameworks as a form of regularization. This approach would serve as an important and strong baseline for sparse view 3D reconstruction method.
6. **Quantitative analysis**: The approach has been validated against a variety of contemporary approaches and state of the art performance is shown on chamfer distance for recovered surface.

**Weaknesses:**

1. **Assumption on 3D information for stage 1**: The losses to train the template prediction network pretrains this network against dataset that have point clouds from COLMAP. Does this limits the applicability mainly to kind of scenes where COLMAP provides enough reconstruction information? Providing some more details about this prior is helpful.
2. **Evaluation**: Although quantitative metrics are provided for geometry, also include image level metrics like PSNR/ SSIM/ LPIPS for the novel view synthesis task is potentially helpful to strengthen the narrative of the evaluation section.
3. **Effect of number of template gaussians**: An ablation study showing how the number of gaussians are chosen and the effect that this number has on the reconstruction quality is instructive.
4. **Additional ablations**: Quantitative ablations are provided for CD as a function of number of input views. However, the manuscript would benefit from the ablations below:
> - *Quantitative ablation* showing the effect of different regularization terms (particularly the SDF constraint and the depth constraints).
> - *Quantitative/ Qualitative ablation* showing the direct optimization of posed sparse views without needing stage 1.
> - *Qualitative ablation* showing the importance of $L_{cov}$ , $L_{radius}$ and $L_{var}$ in Stage 1.
5. **Video Results**: Although not strictly necessary, including turntable video results of the recovered geometry in the supplm will help demonstrate the efficacy of the approach better.


**Questions:**

1. $D_{geo}$ essentially takes the feature map and predicts the center, radius and scale. Does this imply that the template is only based on global level information of the scene, since all local information is lost?
2. The exact specifics of how the spatial resolution of the feature map is flattened for $D_{geo}$ is unclear. Is there a pooling happening between the feature layer and the geometry decoder?

**Limitations:**

Adequate treatment of the limitations of the approach has been provided.

---

> ### Author Rebuttal · Authors · 2023-08-08
>
> Thank you for the encouraging comments.
>
> > **Q1 - Assumption on 3D information for stage 1**
>
> Yes, currently, our template prediction network requires losses with respect to COLMAP reconstructed point cloud for it to learn the surface priors effectively. Hence, the quality of the learned templates largely depends on the quality of the reconstructed point cloud provided for supervision during training, where the predictions of the template network will be impacted if severe holes exist in the point cloud due to texture-less regions. However, we would also like to emphasize that all the methods which make use of COLMAP reconstruction make use of this assumption, e.g., [1].
>
> > **Q2 - Evaluation**
>
> In the below table, we now provide the requested image-level metrics (PSNR/SSIM/LPIPS) for novel view synthesis. Note that MonoSDF only reports PSNR evaluation. As visible, our PSNR outperforms MonoSDF by a significant margin.
>
> |         | PSNR ($\uparrow$) | SSIM ($\uparrow$) | LPIPS ($\downarrow$) |
> |:-------:|:-----------------:|:-----------------:|:--------------------:|
> | MonoSDF |       23.64       |         -         |           -          |
> |   Ours  |     **24.34**     |       0.7208      |         0.264
>
> > **Q3 - Effect of number of templates Gaussians**
>
> We agree that this ablation study is valuable. Due to the required training time for these ablations, it was not feasible to produce them as part of the rebuttal. However, we are currently running them and will add the new results to the revision.
>
> > **Q4 - Additional ablations**
>
> We provide the quantitative evaluation showing the effect of different regularization terms for the object scan 65 shown in the paper. As shown in the table, using both constraints during regularization gives the best result.
>
> |                        | Chamfer Distance (CD) ($\downarrow$) |
> |:----------------------:|:------------------------------------:|
> |   Only SDF Constraint  |                 2.70                 |
> |  Only depth constraint |                 1.96                 |
> | SDF + Depth Constraint |                 **1.52**                 |
>
> We are currently running the remaining two ablations comprising of direct optimization of posed sparse views without needing stage 1 and the importance of loss terms in stage 1. We will add the new results to the revision.
>
> > **Q5 - Video Results**
>
> We will provide a video demonstrating the reconstruction quality of the meshes in the revision.
>
> > **Q6 - Clarifications about $D_{geo}$**
>
> Yes, the templates are based on the global information of the scene since $D_{geo}$, the module responsible for predicting the template parameters, consumes a feature map that encodes the global scene information from the input image. Such a global guidance for sparse-view surface reconstruction is *exactly* the motivation for our work, and Gaussian templates parameterize this guidance. We will clarify this in the revision.
>
> > **Q7 - Clarification about feature mapping in $D_{geo}$**
>
> Yes, it's a kind of pooling achieved by bilinear interpolation. Here we encode the image into feature maps of fixed size, and based on the number of templates, create a uniform grid that is used to interpolate the features bilinearly. In this way, we get the latent codes for each template (aggregated locally), which are then decoded by the $D_{geo}$ and $D_{vox}$ decoders. We will clarify this in the revision.
>
> [1] Xu, Qiangeng, et al. "Point-nerf: Point-based neural radiance fields.", CVPR 2022.

---

### Author Rebuttal · Authors · 2023-08-09

We thank all reviewers for their insightful comments. It is encouraging to see that the reviewers find the addressed problem important (R_4rRP), with a novel (R_jSFe, R_4rRP, R_hUMT) and technically sound (R_bjQu) proposed solution that circumvents the requirement of explicit cues (R_4rRP) and/or 3D supervision (R_Ahim), while achieving high-quality 3D neural surface reconstruction (R_Ahim, R_jSFe).

> **Q1 - Generalizability of the neural template network (TPN) (R#Ahim, R#4rRP, R#bjQu)**

We appreciate the reviewers for raising the issue of generalization to new data distributions. We also realize now that our remark at line 274 in the paper may have cast some doubts about our TPN on this front. This particular remark was made *in principle* since the templates were learned from and characteristic of the training data.

Now we actually test the generalizability of our TPN on a new dataset, namely MobileBrick [1], by comparing the reconstruction results when the neural templates were learned by a pre-trained TPN (on DTU) vs. when they were trained on MobileBrick. Note that overall, the models from these two datasets are quite different in terms of geometry and structure. The last two columns in the Table below and the qualitative results in Figure 2 (see rebuttal PDF) show that the reconstruction qualities under the two scenarios are comparable, attesting to the generalizability of our TPN.

Furthermore, we compare the generalizability of our neural reconstruction framework to that of MonoSDF (original code, with default hyperparameters), under the sparse+disparate input views, tested on MobileBrick. The second column of Table and also Figure 2 in the rebuttal pdf show that the reconstruction results by MonoSDF fall significantly behind those by both versions of our method. We will include these experiments and results in the revision and then revise our remark on line 274 about generalizability accordingly.

Note that we chose MonoSDF for comparison since it is the closest approach to DiViNet in spirit, i.e., both overfit to an input with the reconstruction assisted by a learned prior, and both are applicable to the sparse input setting. However, the priors employed by the two methods are quite different – in the case of MonoSDF, priors come in the form of depth and normal maps, pre-trained on a large-scale dataset with 3D ground-truth supervision, while our work uses Gaussian templates trained on DTU, without requiring any such supervision. In principle, our TPN can benefit from more data and more sophisticated architecture design.

Results demonstrating the generalizability of our method when applied to test models from MobileBricks, with the TPN pre-trained on DTU vs. trained on MobileBrick, also with a comparison to MonoSDF. The metrics used is F1 score as reported by MobileBrick dataset.

|   | MonoSDF | Ours (pre-trained TPN) | Ours (TPN re-training) |
|:---------:|:-------:|:--------------------------:|:--------------------------:|
|   Bridge  |   0.06  |            0.565           |          **0.658**         |
|   Camera  |  0.282  |            0.61            |          **0.67**          |
| Colosseum |  0.055  |            0.219           |          **0.22**          |
|   Castle  |  0.019  |            0.175           |          **0.187**         |

>**Q2 - Reconstruction quality under dense-view scenarios (R#hMUT, R#bjQu)**

Several reviewers pointed out that our method is not the best performer under dense input views, e.g., placing #2 in Table 2 of the main paper. To this, let us state first and foremost that dense-view reconstruction is *not* our focus. More importantly, one probably should not expect that a single method must be the best performer in both sparse *and* dense view settings. We believe that for a method to really excel in one setting, it may rely on specific priors or inductive biases that are different from those necessary in the other setting. For example, multiview consistency is applicable for dense but not disparate views.

Having said the above, we now show, based on reviewer requests, that the performance of our method on dense input views is stronger than what the current results may reflect. First, we compare to GeoNeuS (R#hUMT), which was initially neglected since it was specifically designed for dense-view scenarios. We tested GeoNeuS on MobileBrick and it *underperforms* compared to DiViNet. This demonstrates the robustness of our approach on different datasets, on dense views. The quantitative results are shown in the below table. As per the evaluation protocol of MobileBricks, we use the F1 score ($\uparrow$ is better) to quantify reconstruction accuracy.

|       | Bridge | Camera | Colosseum | Castle |
|:------------------:|:------:|:------:|:---------:|:------:|
|       GeoNeus      |  0.74  |  0.73  |    0.36   |  0.457 |
|        Ours        |  **0.915** |  **0.846** |   **0.415**   |  **0.572** |


Overall, these new results demonstrate that our solution is extendable to dense-view scenarios, without significantly compromising result quality in comparison to SOTA.

[1] Li, Kejie, et al. "MobileBrick: Building LEGO for 3D Reconstruction on Mobile Devices.", CVPR 2023.

---

### Decision · Program_Chairs · 2023-09-21

**Decision:**

Accept (poster)

**Comment:**

This paper was discussed extensively by the authors and reviewers, as well as the reviewers themselves. The final paper had split recommendations from the reviewers. After reading the paper, reviews, rebuttal, and discussion, the area chair is inclined to agree with the accept-leaning reviewers. The AC thanks the reviewers who participated in the discussion and provided feedback to the author, and encourages the authors to very carefully consider and incorporate the reject-inclined reviewers' feedback into the final version of the paper.

Reviewer jSFe championed the paper and summarized the positives of and concerns about the paper well. For the positives, the paper presents work with good novelty and works in an interesting setting (sparse views). On the other hand, the submitted version of the paper was potentially missing baselines and had clarity concerns. In the view of the AC, following jSFe, the authors addressed the concerns of the missing baselines. While huMT disagreed earlier in the discussion, the AC examined the authors' final response in the discussion chain with huMT and believes that the final response does resolve some of the raised issues. The AC is concerned by Ahim's comment about clarity, but thinks that the setting, results, and understanding by the other reviewers overcome the concerns. However, the AC strongly encourages the authors to use Ahim's comments to improve the final version of the paper.